# BEYOND ACCURACY: MEASURING REWARD VARIANCE AS A PREDICTIVE BENCHMARK FOR RLHF

## ABSTRACT

Reward models (RMs) provide the core signal in reinforcement learning from human feedback (RLHF). However, most evaluations focus on pairwise accuracy and overlook how the separability and concentration of reward distributions shape the optimization landscape and convergence rate. To address this limitation, we present the Reward Variance Benchmark (RVB), an evaluation suite that quantifies distributional properties of RM scores. RVB introduces three variance-oriented metrics that capture complementary aspects of an RM's signal: score distribution concentration, global pairwise separation, and cross-prompt decision-style stability. We evaluate 23 widely used RMs on RVB with a toolkit that supports reproducible analysis. On this benchmark, the variance metrics yield stable rankings and, together with accuracy, show preliminary links to downstream convergence behavior in a small-scale RLHF case study. These findings support a key insight: in addition to judging responses correctly, an effective RM in RLHF should also score them with sufficient clarity and stability to provide actionable gradients. Overall, RVB provides a first variance-centric predictive benchmark for analyzing RLHF convergence under a fixed setup, while also supporting more open-ended studies of how reward variance interacts with calibration and policy dependence.

## 1 INTRODUCTION

Reward Models (RMs) guide the alignment of large language models in Reinforcement Learning from Human Feedback (RLHF) (Dong et al., 2023; Ouyang et al., 2022; Bai et al., 2022). By learning from human preferences, an RM assigns a scalar reward to candidate outputs, and its quality largely determines the performance ceiling of the final policy model (Glaese et al., 2022; Wu, 2025; Gao et al., 2023). Beyond policy training, the functions of RMs further extend to data selection (Yuan et al., 2023) and inference (He et al., 2024; Yu et al., 2024). Currently most evaluations focus on **pairwise accuracy**, which measures an RM's ability to distinguish between "chosen" and "rejected" responses (Lambert et al., 2025; Liu et al., 2025). More recent work, such as the RMB (Zhou et al., 2025), adopts a Best-of-$N$ (BoN) setting that better reflects practical selection among multiple candidates and correlates more strongly with downstream performance.

The lower the accuracy, the more prone a reward model is to reward hacking (Skalse et al., 2022; Pang et al., 2023). Yet accuracy is not the whole story (Ivison et al., 2024; Wen et al., 2024). An RM's teaching ability also depends on the variance of its feedback. When a set of candidates receives scores closely concentrated with low variance, the optimization landscape becomes flat, the policy gradient quickly shrinks, and the training process slowly converges (Razin et al., 2025). Thus, in RLHF it is not enough for an RM to rank candidates correctly; it should also assign scores with sufficient separation and stability to provide actionable gradients (Kim et al., 2025). Despite this, the field lacks a systematic, variance-centric evaluation framework that examines RMs from this perspective and connects directly to optimization dynamics. Figure 1 illustrates how identical pairwise accuracy can mask large differences in reward variance and, in turn, the RLHF objective landscape.

We address this gap by proposing the **Reward Variance Benchmark (RVB)**. Building on the task and scenario framework of RMB, RVB constructs a controllable and statistically meaningful BoN candidate pool under fixed sampling controls. The full set covers approximately 1.9k prompts, each associated with 15-17 candidate responses. From this pool, we distill a high quality subset,

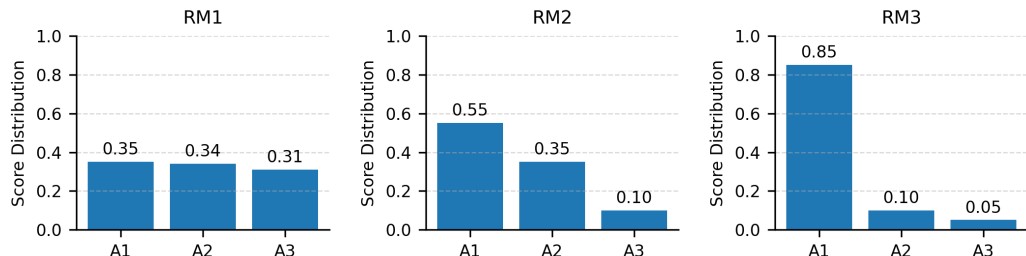

Figure 1: An illustration of why reward variance matters beyond pairwise accuracy. Three reward models (RM1–RM3) rank responses A1–A3 (ordered by ground-truth quality, A1 highest) with identical accuracy, yet yield vastly different score distributions. RM1's flat landscape provides a weak training signal (low variance), while RM3's decisiveness creates a large margin and a strong signal. This highlights that for effective RLHF, RMs must be evaluated on score variance, not just accuracy.

*Eval-Core*, containing 354 prompts with 9 candidates each, enabling fast, reproducible, and apples-to-apples comparisons. Together, Eval-Core and our metrics provide a variance-centric benchmark on which we can systematically study how reward variance relates to RLHF convergence behavior.

To move beyond accuracy, RVB introduces *variance-aware* metrics that jointly profile an RM's reward signal along three complementary axes. First, a *score distribution concentration* measure uses a data-driven temperature to distinguish peaky from flat scoring. Second, a *global pairwise separation* measure aggregates all pairwise gaps, offering robustness to outliers and performing well with small candidate sets. Third, a *cross-prompt decision-style stability* measure is derived from the medians and *interquartile ranges* (IQRs) of the first two. By construction, all three metrics are scale-invariant and rely on robust statistics, and we use them as diagnostic axes for scoring and comparing RMs on RVB rather than as training objectives.

Our evaluation of 23 prominent reward models, together with small-scale RLHF experiments on a subset of them, suggests that different variance profiles are associated with systematic differences in convergence behavior, and provide a first variance-aware view of RMs.

Our main contributions are as follows:

- **Variance-centric benchmark and dataset**: We build RVB on top of RMB and distill *Eval-Core*, a controlled Best-of-$N$ subset (354 prompts with 9 responses each) with standardized score matrices and tools for variance-centric analysis of reward models in RLHF.

- **Variance-oriented metric suite**: We introduce three robust, scale-invariant metrics: **SEI** for score distribution concentration, **nGMD** for global pairwise separation, and **DCI** for cross-prompt decision stability, and relate them to simple baselines such as variance, entropy, and Brier score.

- **Empirical analysis of 23 reward models**: We profile 23 public RMs with these metrics, group them into variance-based "teaching styles", and show through GRPO case studies how different profiles manifest as distinct convergence behaviors.

- **Methodological insights and open directions**: We analyze how reward variance interacts with accuracy, calibration, and policy choice, and highlight open questions about what constitutes a better reward model across tasks and objectives.

## 2 BACKGROUND

### 2.1 THE EVOLUTION OF REWARD MODEL EVALUATION: FROM ACCURACY TO SCENARIOS

Evaluating reward models (RMs) is central to understanding and improving LLM alignment. We follow the taxonomy of Lambert et al. (2025), in which RM mechanisms include discriminative reward (Cai et al., 2024), generative reward (Zheng et al., 2023), and implicit reward (Rafailov

et al., 2023). At different granularities, RMs are grouped into outcome-based reward models (ORMs) (Cobbe et al., 2021), which score final outputs, and process-based reward models (PRMs) (Lightman et al., 2023), which assess intermediate reasoning steps but require costly step-wise annotations. Our work focuses on ORMs.

To standardize RM evaluation, the community has developed a series of benchmarks that reflect an evolving understanding of RM capabilities.

**Accuracy-Centric ORM Benchmarks**. RewardBench (Lambert et al., 2025) establishes pairwise accuracy on prompt–chosen–rejected triplets as a core paradigm and maintains a widely used leaderboard. Subsequent benchmarks probe robustness, text transformations, mathematical reasoning, multilinguality, RAG, and proxy tasks, including RM-Bench (Liu et al., 2025), reWord-Bench (Wu et al., 2025), rewardMATH (Kim et al., 2024), M-RewardBench (Gureja et al., 2025), RAG-RewardBench (Jin et al., 2025), and PPE (Frick et al., 2024).

**Scenario-Aware and Best-of-N Evaluation**. To better approximate deployment, RMB (Zhou et al., 2025) evaluates how an RM selects the best response among $N$ candidates in a Best-of-$N$ setting, which aligns with inference-time reranking and preference-data generation and correlates more strongly with downstream alignment than pairwise accuracy; RewardBench 2 (Malik et al., 2025) and RBoN (Ichihara et al., 2025) further refine this paradigm with multi-skill evaluation and regularised sampling. Beyond LLM-specific text benchmarks, TAC (Muslimani et al., 2025), EPIC (Gleave et al., 2020), and DARD (Wulfe et al., 2022) study learned reward functions in control environments at the trajectory or transition level, in ways that are invariant to potential-based shaping and predictive of policy quality; RVB instead profiles the distributional and variance behaviour of outcome-based text reward models in a fixed Best-of-$N$ setting and is complementary to these control-oriented frameworks.

## 2.2 A SHIFT IN PERSPECTIVE: DECOUPLING ACCURACY FROM TEACHING EFFECTIVENESS

Although these benchmarks measure ordering quality, several studies show that ordering accuracy and teaching effectiveness can diverge.

**Empirical Evidence**. Chen et al. (2024) demonstrates an accuracy paradox, where training with a moderately accurate RM can outperform training with the most accurate RM. Related work (Leng et al., 2024) reports overconfidence and poor calibration in many RMs. Absolute scores and their gaps, not only ranks, can transmit uncalibrated signals to the policy and lead to undesirable behaviors. These observations motivate a direct analysis of the reward distribution.

**Theoretical Interpretation**. From an optimization view, Razin et al. (2025) shows that reward variance is a key driver of RLHF efficiency. Low variance flattens the optimization landscape, weakens policy gradients, and slows learning, irrespective of ranking accuracy. This interpretation explains the accuracy paradox and suggests that a more discriminative RM with higher variance can act as a more effective teacher even if its accuracy is slightly lower.

## 2.3 VARIANCE IN PRACTICE AND THE REMAINING RESEARCH GAP

Recent work has started to treat reward variance as a first-class design target in both optimization and modeling. On the algorithmic side, DGRO (Su et al., 2025), GVPO (Zhang et al., 2025), and GRPOVI (Yang et al., 2025b) explicitly adjust reward variance during RLHF training, either by reshaping the objective or by controlling the trade-off between exploration and exploitation, with the goal of improving convergence stability and sample efficiency. In parallel, distributional and uncertainty-aware reward models, such as URM (Lou et al., 2024) and quantile-regression reward models like QRM (Dorka, 2024), aim to regularize or model the spread of reward scores through ensembles or predictive distributions instead of a single deterministic scalar.

Despite these advances, there is still no standardized, model-agnostic benchmark for measuring and comparing the variance behaviour of pre-trained reward models themselves. RVB is designed to fill this gap: given any reward model, including those trained with DGRO, GVPO, GRPOVI or ensemble-based and calibrated variants, it provides a common Eval-Core dataset and variance-aware metrics that profile the concentration, global separation, and cross-prompt stability of the teaching signals they produce.

# 3 BENCHMARK CONSTRUCTION

## 3.1 BASE CORPUS AND CANDIDATE POOL CONSTRUCTION

We build our candidate pool from the helpfulness subset of RMB (Zhou et al., 2025), which covers 12 task categories and 37 real-world scenarios, and adopts the Best-of-$N$ (BoN) paradigm.

Our candidate pool covers 1,906 prompts. For each prompt, we keep the RMB candidates (about 3–6) and add GPT-4o (Hurst et al., 2024) generations at four temperatures (0.2, 0.7, 1.0, 1.3), three per temperature targeted at low, medium, and high quality; in total this yields 29,148 responses, about 15–17 candidates per prompt and a controlled quality gradient. We apply light filtering of malformed or degenerate outputs. Prompt templates used for data generation are detailed in Appendix A.7. While our main experiments use GPT-4o as the generator, Appendix A.4.2 details a protocol for instantiating RVB with alternative generators and reports a small-scale variant based on a Qwen2-family model, where we compare the resulting RVB rankings against the GPT-4o-based leaderboard. This richer candidate set supports stable estimation of dispersion statistics such as the Gini mean difference and interquartile range.

## 3.2 SCORE COLLECTION AND SCALE UNIFICATION

We use a standardized pipeline for scoring and normalization so that scores from different RMs can be put on a common scale.

**RM Scoring**. We score the full candidate pool with three representative RMs: *OpenAssistant/reward-model-deberta-v3-large-v2* (OpenAssistant), *nicolinho/QRM-Llama3.1-8B-v2* (Dorka, 2024), and *CIR-AMS/BTRM_Qwen2_7b_0613* (CIR-AMS).

**Two-Track Normalization**: Since raw scores are not directly comparable across RMs, we apply two complementary procedures.

*(i) Percentile-rank normalization for sampling.* During sampling for *Eval-Core*, we applied an independent percentile-rank transform to each RM's scores (Roscoe, 1969), mapping them to $[0, 1]$. This removes differences in absolute scale, makes relative rankings comparable across models, and enables reliable inter-model consensus estimates.

*(ii) Robust standardization for reporting.* For final metric calculation and reporting, we preserve relative scale while mitigating the effects of outliers by employing a robust standardization procedure. For each RM $m$, we recenter its scores $r$ by the median of its full-set scores ($\mathcal{R}_m$) and scale them by a robust global factor $s_{\text{RM}}^{(m)}$, calculated as $1.4826\times$ Median Absolute Deviation (MAD) (Ruppert & Matteson, 2011):

$$\text{MAD}(\mathcal{R}_m) = \text{median}_{r \in \mathcal{R}_m}\left| r - \text{median}(\mathcal{R}_m) \right|, \qquad s_{\text{RM}}^{(m)} := 1.4826 \cdot \text{MAD}(\mathcal{R}_m), \quad (1)$$

where the constant 1.4826 calibrates this scale factor to be comparable to a standard deviation under normality. This process yields **robustly standardized scores** ($z$), a robust version of the standard $z$-score, for each candidate response $i$ to a prompt $p$ from a model $m$:

$$z_{i,m}^{(p)} := \frac{r_{i,m}^{(p)} - \text{median}(\mathcal{R}_m)}{s_{\text{RM}}^{(m)}}, \quad (2)$$

where $r_{i,m}^{(p)}$ is the raw score for candidate $i$ of prompt $p$ from model $m$. All publicly reported variance and separation metrics use these standardized scores to ensure fair cross-model comparison.

## 3.3 EVAL-CORE: A TWO-STAGE SAMPLING PROCESS BASED ON DISCRIMINATION AND CONSENSUS

To release a public subset that is both representative and reproducible, we construct *Eval-Core* via a two-stage stratified sampling process.

**Stage 1: Prompt selection by separation and consensus.** Using percentile-ranked scores, we compute two prompt-level meta-metrics: a Relative Spread Index (RSI, the average 90th–10th percentile span) and an inter-RM Spearman consensus. We restrict to prompts in the top 35% of RSI and, within this high-span set, sample a balanced mixture of high- and low-consensus prompts.

**Stage 2: Candidate selection ensuring quality gradient and diversity.**

For each selected prompt, we choose nine candidates by splitting average percentile ranks at the 20th and 80th percentiles into low, medium, and high tiers and sampling three candidates per tier while encouraging diversity in model sources and temperature settings. This yields an *Eval-Core* subset of 354 prompts with 3,186 responses in total. Eval-Core, together with our variance-oriented metrics, thus forms a benchmark where reward models can be scored, ranked, and compared in terms of both accuracy and variance-related properties, with downstream RLHF convergence used as a reference task in our case study.

## 4 METRIC SUITE

To comprehensively and robustly characterize reward-model score distributions, we design a suite of five complementary metrics and, after a correlation analysis, retain three as the primary set. See Appendix A.2 for details on the two remaining metrics and the correlation study.

The retained metrics are **SEI** (Softmax–Entropy Index) to quantify score distribution concentration, **nGMD** (normalized Gini mean difference) for global pairwise separation, and **DCI** (Decision Consistency Index) for cross-prompt decision-style stability. Together, these metrics profile an RM's "teaching signal": how it concentrates scores, separates candidates, and maintains consistency. Our design follows two principles: *robustness*, using outlier-resistant statistics (e.g., median, IQR), and *interpretability*, favoring quantities with a clear qualitative meaning over raw variance-like moments. In RVB, SEI, nGMD, and DCI act as the axes along which reward models are scored and ranked on Eval-Core, providing a structured view of concentration, separation, and stability that is predictive of GRPO-style RLHF convergence in our case study. We do not, however, propose these metrics as training objectives.

### 4.1 UNIFIED SCALE AND AGGREGATION

All metrics in this section (e.g., SEI, nGMD) are computed on the robustly standardized scores $z$ defined in Sec. 3.2 (Eq. 2), which remove scale and offset differences between reward models via the global scale factor $s_{\mathrm{RM}}^{(m)}$ (Eq. 1). For each metric, we first compute a per-prompt value and then take the median across prompts as the RM-level score (suffix $\cdot_{\mathrm{med}}$), which is robust to outlier prompts and works well for small candidate pools (e.g., $n = 9$). Task-normalized variants are reported in Appendix A.4.1.

### 4.2 $\mathrm{SEI}_{\mathrm{med}}$: SCORE CONFIDENCE CONCENTRATION

**Definition (Softmax–Entropy Index, SEI).** For each prompt $p$, we define $\mathrm{SEI}(p)$ as

$$\mathrm{SEI}(p) \;=\; 1 \;-\; \frac{H(\mathbf{p})}{\log n}, \quad \text{where} \quad \mathbf{p} \;=\; \mathrm{softmax}\!\left(\frac{\mathbf{r}}{\tau_p}\right), \quad \tau_p \;=\; \frac{\mathrm{IQR}_p}{1.349}\,. \tag{3}$$

Here, $H(\cdot)$ is the Shannon entropy in nats and $\mathbf{r} \in \mathbb{R}^n$ is the vector of $n$ reward scores for the $n$ candidate responses of prompt $p$. The adaptive temperature $\tau_p$ is set such that $\tau_p \approx \sigma$ under a normal assumption (since $\mathrm{IQR} \approx 1.349\sigma$), which removes a manual hyperparameter. Because $\frac{H(\mathbf{p})}{\log n} \in [0, 1]$, we have $\mathrm{SEI}(p) \in [0, 1]$. Finally, $\mathrm{SEI}_{\mathrm{med}}$ is the median of $\mathrm{SEI}(p)$ over all prompts:

$$\mathrm{SEI}_{\mathrm{med}} \;=\; \mathrm{median}_p\, \mathrm{SEI}(p).$$

**Semantics and motivation**. SEI is a normalized entropy index that measures how concentrated an RM's scores are for a fixed candidate set. High $\mathrm{SEI}_{\mathrm{med}}$ (close to 1) means a peaky, decisive distribution that clearly favors a few candidates, while low $\mathrm{SEI}_{\mathrm{med}}$ (close to 0) means a flat, uncertain distribution over many candidates. In RVB, we use $\mathrm{SEI}_{\mathrm{med}}$ as the concentration axis for comparing reward models.

### 4.3 $\text{nGMD}_{\text{med}}$: GLOBAL PAIRWISE SEPARATION

**Definition (normalized Gini mean difference, nGMD).** For each prompt $p$, we define $\text{nGMD}(p)$ as

$$\text{nGMD}(p) = \frac{\frac{2}{n(n-1)} \sum_{1 \le i < j \le n} |r_i - r_j|}{s_{\text{RM}}^{(m)}}, \tag{4}$$

where $r_i, r_j \in \mathbf{r}$ are reward scores. $s_{\text{RM}}$ is the robust global factor defined in equation 1. $\text{nGMD}_{\text{med}}$ is defined as the median of $\text{nGMD}(p)$ over all prompts: $\text{nGMD}_{\text{med}} = \text{median}_p \, \text{nGMD}(p)$ .

**Semantics and motivation**. GMD aggregates all pairwise gaps, so it captures overall separation more comprehensively than a top-gap metric. Compared with variance, it is more robust to outliers and behaves well for small candidate sets (e.g., 5–10), so a larger $\text{nGMD}_{\text{med}}$ means the RM consistently induces substantial spread among candidates. Within RVB, we use $\text{nGMD}_{\text{med}}$ as the global separation axis on Eval-Core.

### 4.4 DCI: DECISION STYLE STABILITY

**Definition (RM-level, across prompts).**

$$S_{\text{nGMD}} = \big\{\text{nGMD}(p)\big\}_{p \in P}, \qquad S_{\text{SEI}} = \big\{\text{SEI}(p)\big\}_{p \in P}. \tag{5}$$

where $P$ is the set of prompts. For each cross-prompt sequence, compute a robust level-to-dispersion ratio using median/IQR:

$$D_{\text{nGMD}} = \frac{\text{median}\big(S_{\text{nGMD}}\big) + \varepsilon}{\max\big(\text{IQR}\big(S_{\text{nGMD}}\big), \delta\big)}, \qquad D_{\text{SEI}} = \frac{\text{median}\big(S_{\text{SEI}}\big) + \varepsilon}{\max\big(\text{IQR}\big(S_{\text{SEI}}\big), \delta\big)}. \tag{6}$$

Here $\varepsilon$ avoids near-zero numerators and $\delta$ stabilizes tiny IQRs. The final index is

$$\text{DCI} = \exp\left(-\frac{\kappa}{D_{\text{nGMD}} + D_{\text{SEI}}}\right) \in (0, 1] \tag{7}$$

with a scale constant $\kappa > 0$ (default $\kappa = 2$). When both $D_{nGMD}$ and $D_{SEI}$ are large and stable, DCI approaches 1; if either sequence varies strongly across prompts, DCI decreases.

**Semantics and motivation**. DCI captures whether an RM maintains consistent separation and score concentration across prompts. High DCI (close to 1) reflects a coherent, predictable evaluation style, while low DCI indicates more opportunistic behavior, so DCI complements $\text{SEI}_{\text{med}}$ and $\text{nGMD}_{\text{med}}$ by summarizing cross-prompt stability rather than per-prompt levels.

### 4.5 COMPOSITE SCORE

To provide a single coarse ranking for convenience, we also define a composite score by computing MAD-$z$ scores across RMs for each metric in $\text{SEI}_{\text{med}}$, $\text{nGMD}_{\text{med}}$, and DCI, and averaging them with equal weights:

$$\text{Composite}_j = \sum_{m \in \{\text{SEI}_{\text{med}}, \, \text{nGMD}_{\text{med}}, \, \text{DCI}\}} \frac{m_j - \text{median}_k(m_k)}{\text{MAD}_k(m_k)}, \tag{8}$$

where $j$ indexes a specific reward model and $k$ ranges over all RMs.

Together, the three metrics cover score distribution concentration (SEI), global pairwise separation (nGMD), and cross–prompt decision–style stability (DCI). The composite score encourages balanced performance across variance dimensions and avoids over-reliance on any single metric. We treat this composite as an optional summary index for the benchmark rather than a primary object of interpretation or optimization: our analyses focus on the three axes $\{\text{SEI}_{\text{med}}, \text{nGMD}_{\text{med}}, \text{DCI}\}$, and we show in Tab. 3 in App. A.3 that alternative weightings (e.g., PCA loadings) yield highly similar rankings.

**Metric relationships and predictive signal**. To understand how the three RVB metrics relate to accuracy, we run a PCA over four standardized features across all 23 RMs: RewardBench accuracy,

Table 1: RVB leaderboard across 23 reward models. Entries are ordered by the Composite Score (higher is better). For each model we report $\text{SEI}_{\text{med}}$, $\text{nGMD}_{\text{med}}$, and DCI with ranks in parentheses, and the RewardBench Score (pairwise accuracy, %). "N/A" indicates no public RewardBench result as of September 20, 2025. Best values per column are shown in bold.

| Model Name | Composite Score | $\text{SEI}_{\text{med}}$ (Rank) | $\text{nGMD}_{\text{med}}$ (Rank) | DCI (Rank) | RewardBench Score |
|---|---|---|---|---|---|
| URM-LLaMa-3.1-8B | **2.21** | 0.141 ( 9 ) | 1.086 ( 3 ) | **0.856 ( 1 )** | 92.9 |
| Skywork-Reward-Gemma-2-27B | 1.69 | 0.197 ( 4 ) | 1.090 ( 2 ) | 0.593 ( 11 ) | 93.8 |
| QRM-Gemma-2-27B | 1.51 | 0.184 ( 5 ) | 1.081 ( 4 ) | 0.603 ( 9 ) | **94.4** |
| Skywork-Reward-Llama-3.1-8B | 1.23 | **0.279 ( 1 )** | 1.020 ( 5 ) | 0.379 ( 22 ) | 92.5 |
| Skywork-Reward-Llama-3.1-8B-v0.2 | 0.95 | 0.252 ( 2 ) | 0.985 ( 7 ) | 0.392 ( 21 ) | 93.1 |
| GRM-Llama3-8B-rewardmodel-ft | 0.42 | 0.228 ( 3 ) | 0.999 ( 6 ) | 0.344 ( 23 ) | 91.5 |
| beaver-7b-v2.0-reward | 0.11 | 0.082 ( 23 ) | **1.094 ( 1 )** | 0.609 ( 8 ) | 63.9 |
| Skywork-Reward-V2-Qwen3-1.7B | 0.09 | 0.128 ( 12 ) | 0.969 ( 8 ) | 0.592 ( 14 ) | N/A |
| RM-Mistral-7B | 0.00 | 0.109 ( 22 ) | 0.956 ( 11 ) | 0.640 ( 4 ) | 80.4 |
| Skywork-Reward-V2-Llama-3.2-3B | -0.06 | 0.136 ( 10 ) | 0.967 ( 9 ) | 0.542 ( 19 ) | N/A |
| Gemma-2B-rewardmodel-baseline | -0.09 | 0.119 ( 16 ) | 0.959 ( 10 ) | 0.592 ( 13 ) | 73.4 |
| Llama-3.1-Tulu-3-8B-DPO-RM-RB2 | -0.09 | 0.118 ( 18 ) | 0.913 ( 17 ) | 0.635 ( 5 ) | 84.3 |
| Llama-3.1-Tulu-3-8B-RL-RM-RB2 | -0.09 | 0.120 ( 15 ) | 0.922 ( 15 ) | 0.622 ( 7 ) | 83.7 |
| BTRM_Qwen2_7b_0613 | -0.11 | 0.110 ( 21 ) | 0.915 ( 16 ) | 0.651 ( 2 ) | 83.2 |
| Llama-3.1-Tulu-3-8B-SFT-RM-RB2 | -0.11 | 0.126 ( 14 ) | 0.927 ( 13 ) | 0.596 ( 10 ) | 85.5 |
| Skywork-Reward-V2-Qwen3-8B | -0.11 | 0.136 ( 11 ) | 0.949 ( 12 ) | 0.548 ( 17 ) | N/A |
| Skywork-Reward-V2-Qwen3-4B | -0.18 | 0.126 ( 13 ) | 0.923 ( 14 ) | 0.586 ( 15 ) | N/A |
| QRM-Llama3.1-8B-v2 | -0.18 | 0.166 ( 6 ) | 0.843 ( 20 ) | 0.546 ( 18 ) | 93.1 |
| Llama-3.1-8B-Instruct-RM-RB2 | -0.27 | 0.112 ( 20 ) | 0.902 ( 18 ) | 0.628 ( 6 ) | 88.8 |
| Llama-3.1-8B-Base-RM-RB2 | -0.38 | 0.119 ( 17 ) | 0.898 ( 19 ) | 0.592 ( 12 ) | 84.6 |
| RAMO-Llama3.1-8B | -0.71 | 0.155 ( 7 ) | 0.819 ( 21 ) | 0.499 ( 20 ) | N/A |
| tulu-v2.5-13b-uf-rm | -0.80 | 0.113 ( 19 ) | 0.772 ( 22 ) | 0.642 ( 3 ) | 45.7 |
| ArmoRM-Llama3-8B-v0.1 | -0.82 | 0.143 ( 8 ) | 0.737 ( 23 ) | 0.585 ( 16 ) | 90.4 |

$\text{SEI}_{\text{med}}$, $\text{nGMD}_{\text{med}}$, and DCI. The first three principal components already explain almost all variance in this 4D space and show a clear structure (Tab. 3 in App. A.3): PC1 is dominated by accuracy and SEI, PC2 is driven by nGMD, and PC3 mainly reflects a joint direction of accuracy and DCI. This matches the design goal: SEI partly overlaps with accuracy along a "sharpness" axis, while nGMD and DCI contribute complementary dispersion and stability directions that are not captured by accuracy alone. In RVB, these three dimensions serve as the benchmark axes along which reward models are scored and compared on Eval-Core.

## 5 EVALUATION

We evaluate 23 popular reward models on RVB. The resulting scores define a variance-oriented leaderboard and reveal distinct patterns in "teaching signals" that accuracy metrics alone do not capture. For comparison, we also report each model's official RewardBench score, allowing for a direct contrast between accuracy-centric and variance-centric views. We use this leaderboard as a tool for analyzing and comparing reward models, rather than as a universal ranking of "best teachers" for all downstream settings.

We further introduce *Eval-Semantic*, a small auxiliary benchmark that tests whether models assign different scores to semantically equivalent statements (see Appendix A.6 for design and results).

### 5.1 RVB LEADERBOARD AND OVERALL RESULTS

Table 1 summarizes RVB performance for all 23 RMs, ranking them by our composite score alongside their official RewardBench accuracy.

**Accuracy vs. variance.** For models with public RewardBench scores, RVB composite scores show a moderate positive association with pairwise accuracy (Pearson $r \approx 0.51$, Spearman $\rho \approx 0.48$), indicating partial alignment. However, there are substantial exceptions. For example, QRM-Llama3.1-8B-v2 and ArmoRM-Llama3-8B-v0.1 achieve high accuracy yet rank low on RVB, while beaver-7b-v2.0-reward and tulu-v2.5-13b-uf-rm show strong variance-side signals despite modest or low accuracy. In short, accuracy reflects ordering correctness, whereas RVB captures the strength and stability of the training signal. We therefore view accuracy and RVB metrics as complementary: the leaderboard is useful for analyzing and contrasting reward models that may look similar by accuracy alone, rather than for prescribing a single "best" model for all RLHF use cases. Notably, our model suite already contains several variance-aware or uncertainty-aware teachers, such as URM-LLaMa-3.1-8B and QRM-Llama3.1-8B-v2, which model reward distributions or predictive uncer-

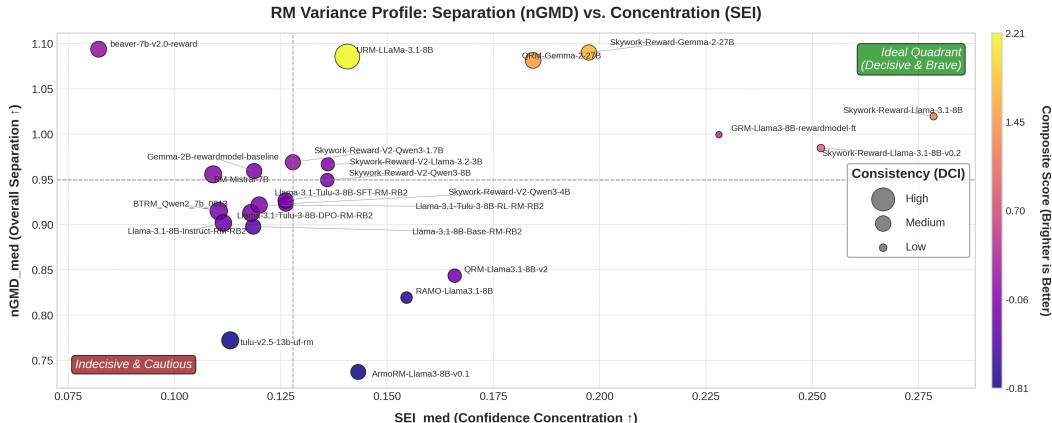

Figure 2: **SEI vs. nGMD** scatter. X-axis: $\text{SEI}_{med}$ (confidence concentration); Y-axis: $\text{nGMD}_{med}$ (global pairwise separation). Point size encodes DCI (decision-style stability) and color encodes the composite RVB score. Only a few models, notably URM-LLaMa-3.1-8B, lie in the top-right region with large, bright points, while beaver-7b-v2.0-reward and Skywork-Reward-Llama-3.1-8B form a contrasting pair, illustrating that concentration, separation, and stability are partially decoupled and require a multi-axis view.

tainty rather than a single point estimate. RVB places these distributional or ensemble-style reward models on the same footing as conventional RMs, revealing that explicit variance regularization at training time does not automatically yield a favorable ex-post variance profile.

## 5.2 FOUR VARIANCE PROFILES AS CONCEPTUAL "TEACHING STYLES"

We use the RVB metrics to describe four representative variance profiles as conceptual "teaching styles". These labels refer to regions of the leaderboard (e.g., high concentration and high stability vs. low concentration and low stability), rather than formal clusters or design prescriptions.

**Consistent and separating**: URM-LLaMa-3.1-8B has strong separation, high decision consistency, and mid-to-high concentration, yielding clean and reliable reward signals.

**Strong but unstable**: Skywork-Reward-Llama-3.1-8B, and to a lesser extent Skywork-Reward-Gemma-2-27B, show very high concentration and competitive separation but low cross-prompt stability, producing sharp yet more volatile gradients.

**Broadly discriminative but indecisive**: beaver-7b-v2.0-reward achieves strong global separation but low concentration and only moderate stability, often distinguishing many candidates without a clear winner.

**Conservative and cautious**: tulu-v2.5-13b-uf-rm and BTRM_Qwen2_7b_0613 combine high cross-prompt stability with weaker separation and concentration, giving steady but relatively weak optimization signals.

Overall, we observe a trade-off between stability and signal strength: some models are sharply discriminative but unstable, while others are stably conservative, and models that balance concentration, separation, and stability tend to rank higher on the composite leaderboard.

## 5.3 METRIC COMPLEMENTARITY AND VISUALIZATION

Our three core metrics capture complementary aspects of variance. Figure 2 visualizes these trade-offs by plotting $\text{SEI}_{med}$ (belief concentration) on the x-axis and $\text{nGMD}_{med}$ (global separation) on the y-axis; point size encodes DCI (cross-prompt consistency) and color encodes the composite score.

Few models reach the upper-right region with large points (high concentration, high separation, high stability). URM-LLaMa-3.1-8B stands out by combining top-tier nGMD with the best DCI. Meanwhile, clear *metric decoupling* also appears: beaver-7b-v2.0-reward sits upper left (high separation,

Table 2: **Offline BoN behavioral fingerprints.** A Qwen2.5-1.5B SFT policy samples $N=4$ candidates per prompt, and each RM selects the best-of-$N$ response. The table reports average answer length on 100 general prompts (Anthropic HH) and refusal rate on 100 safety prompts (BeaverTails).

| Reward model | Length on general prompts | Refusal rate on safety prompts |
|---|---|---|
| Beaver-7B-v2.0-RM | $981 \pm 429$ chars | 17% (17/100) |
| URM-LLaMa-3.1-8B | $1189 \pm 231$ chars | 14% (14/100) |
| Skywork-Reward-Llama-3.1-8B | $1232 \pm 193$ chars | 14% (14/100) |

low concentration), whereas Skywork-Reward-Llama-3.1-8B moves rightward (high concentration, moderate separation) but exhibits a small point size (low DCI). These patterns reinforce the need for a multi-metric view rather than reliance on any single axis.

Beyond these global patterns, we briefly probe task-wise behavior, normalization controls, and generator choice (Appendix A.4.1, A.4.2). A task-level heatmap (Appendix Figure 5) shows that several reward models have clear task-specific strengths; for example, QRM-Llama3.1-8B-v2 looks particularly strong on paraphrasing-style tasks despite a lower overall composite, which suggests room for mixed or curriculum-style training. When we recompute metrics with task-wise normalization (normalizing within each task before aggregation), rankings based on nGMD and DCI stay largely stable, whereas SEI is more sensitive to task mix, so concentration should be read with the task distribution in mind. A small multi-generator check on a Qwen2-7B-Instruct candidate pool for a subset of prompts yields RVB rankings that still match the GPT-4o-based leaderboard at the level of coarse tiers, but with some local rank swaps, indicating a mild dependence on the underlying generator.

### 5.4 BEHAVIOR-LEVEL ANALYSIS

To connect RVB profiles more directly to model behavior, we run an offline Best-of-$N$ (BoN) study: a fixed Qwen2.5-1.5B SFT policy samples $N=4$ responses for 100 general prompts from Anthropic HH and 100 safety prompts from BeaverTails, and three RMs from our RL study (Beaver-7B-v2.0-RM, URM-LLaMa-3.1-8B, Skywork-Reward-Llama-3.1-8B) select the BoN response. We compare answer length on general prompts and refusal rate on safety prompts, and observe that the two RMs with higher SEI and DCI (URM and Skywork) tend to select longer and more length-stable answers than Beaver, while showing slightly lower refusal rates. In other words, sharper and more stable RMs behave like more assertive teachers that push the policy toward longer, more elaborated answers, whereas the more moderate-variance RM behaves more conservatively and refuses slightly more often. These behavior-level patterns are consistent with the "teaching style" profiles in Section 5.3, and detailed statistics are reported in Tab. 2.

## 6 VERIFICATION

### 6.1 RLHF TRAINING DYNAMICS

We evaluate the predictive value of RVB with a small RLHF case study, as shown in Figure 3. The policy is Qwen2.5-1.5B-Instruct (Qwen et al., 2025) trained with GRPO (Shao et al., 2024) on the Anthropic/hh-rlhf dataset. We select three reward models from the leaderboard—URM-LLaMa-3.1-8B (Lou et al., 2024), Skywork-Reward-Llama-3.1-8B (Liu et al., 2024), and Llama-3.1-Tulu-3-8B-DPO-RM-RB2 (Malik et al., 2025)—and track the trajectory of average reward (reward_mean) across training steps.

**Skywork-Reward-Llama-3.1-8B** achieves the highest SEI score and delivers the most concentrated reward signals. Training with this RM yields a clear reward increase after roughly 300 steps.

**URM-LLaMa-3.1-8B** begins to improve rewards at about 330 steps. Although it starts later, once stabilized its reward curve fluctuates less than that of the Skywork model.

**Llama-3.1-Tulu-3-8B-DPO-RM-RB2** produces relatively conservative signals, with rewards staying close to the initialization until around step 500. Even by step 800, its average reward remains below the other two models.

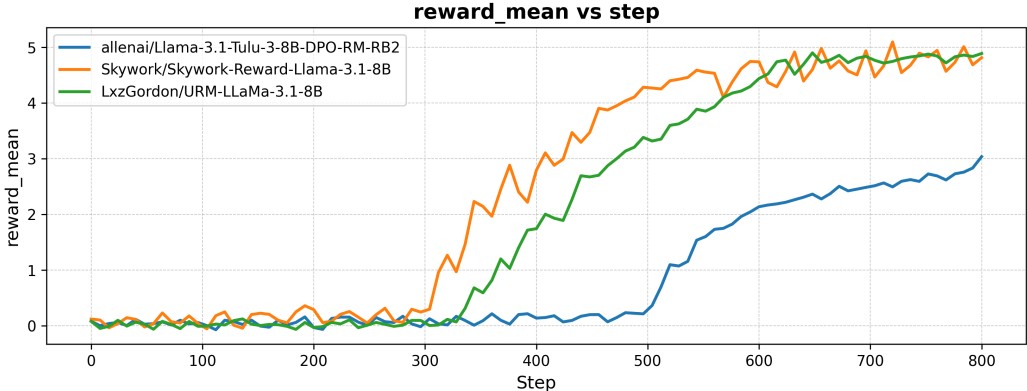

Figure 3: Average reward (reward_mean) vs. training steps on Anthropic HH-RLHF under GRPO with a Qwen2.5-1.5B policy. We compare three reward models from the RVB leaderboard: Skywork-Reward-Llama-3.1-8B, URM-LLaMa-3.1-8B, and Llama-3.1-Tulu-3-8B-DPO-RM-RB2. Higher curves indicate faster reward improvement in this setup.

Taken together, these curves suggest that higher RVB composite scores are associated with faster reward improvement in this setup. The case study also shows how the variance axes help explain the different learning dynamics of URM-LLaMa-3.1-8B, Skywork-Reward-Llama-3.1-8B, and the more conservative Tulu reward model, even when accuracy is similar. However, this experiment is small in scale, so it should not be used to make general claims about universally better reward-model "teachers".

### 6.2 REGRESSION ANALYSIS WITH VARIANCE BASELINES

We run an RM-level regression over the 23 models, relating RewardBench accuracy and variance features to PPO reward AUC on Anthropic/hh-rlhf under the setup in Sec. 6.1. For each RM, the target is the area under its reward–step curve. Accuracy alone explains about one third of the variance in reward AUC (adjusted $R^2 \approx 0.33$). Adding either the three RVB metrics ($\text{SEI}_{\text{med}}$, $\text{nGMD}_{\text{med}}$, DCI) or simple variance baselines (per-prompt normalized standard deviation, prediction entropy, Brier score) raises adjusted $R^2$ to about $0.55$, and combining all features does not improve it further because of multicollinearity. Variance therefore carries useful signal for PPO reward AUC, but much of the linear structure is already captured by simple dispersion statistics; RVB mostly re-expresses this signal along axes of concentration, separation, and stability. Given the small sample, strong collinearity, and short non-converged runs, we view these regressions as evidence that RVB relates to early-stage training behaviour in this PPO setting, not as a universal ordering of reward models (details in App. A.5).

## 7 CONCLUSION

Accuracy-centric evaluation of reward models overlooks their effectiveness as instructors in RLHF. We instead treat reward variance as a critical teaching signal that shapes optimization and training dynamics, and operationalize this view with RVB, a variance-centric evaluation suite and dataset that systematically quantifies concentration, separation, and cross-prompt stability of outcome-based reward models. On our Eval-Core benchmark with a PPO-based RLHF setup, RVB scores provide a predictive benchmark for reward AUC and explain more variance than RewardBench accuracy alone. However, our study is limited to a single helpfulness dataset, one policy family, and short-horizon PPO runs without clear convergence, so RVB should not be interpreted as a universal teacher ranking across tasks, policies, or safety regimes. We view RVB as both a practical benchmark for studying reward models as teachers in our setting and a measurement tool that offers shared data and variance-aware axes for future work on safety, calibration, and policy-dependent notions of "better" reward models.

## ETHICS STATEMENT

Our research adheres to the ICLR Code of Ethics. The primary goal of this work is to introduce a new evaluation dimension for reward models (RMs) that focuses on their "teaching effectiveness" in Reinforcement Learning from Human Feedback (RLHF). By developing the Reward Variance Benchmark (RVB), we aim to foster the creation of more efficient and robust AI alignment methods, which is a positive contribution to AI safety. The construction of our benchmark dataset, Eval-Core, is based on established and publicly available datasets like the RMB (Zhou et al., 2025) , supplemented by controlled generations from GPT-4o. Our two-stage sampling process was designed to be transparent and to ensure a diverse set of responses with a clear quality gradient, mitigating potential biases that could arise from a single data source. We believe our research provides valuable tools and insights for the community without introducing new risks of misuse, as it is focused on evaluation rather than the generation of potentially harmful content.

## REPRODUCIBILITY STATEMENT

We provide all the implementation and methodological details needed to reproduce our results in Sec. 3, 4, and 5. Upon acceptance, we will publicly release the full codebase, datasets, and experiment configurations, including the complete Eval-Core benchmark subset, the source code to compute our variance-sensitive metrics, the collected score matrices from all 23 evaluated reward models, and the scripts and configuration files used for the RLHF verification experiments that assess correlations between our metrics and training convergence. Our benchmark construction pipeline—candidate pool generation (Section 3.1), score normalization (Section 3.2), and stratified sampling for Eval-Core (Section 3.3)—is described in detail to ensure methodological transparency. All experiments use open-source frameworks and models, which are properly cited and accompanied by links in the references.

## THE USE OF LARGE LANGUAGE MODELS (LLMs)

In the preparation of this manuscript, we utilized Large Language Models (LLMs) as assistive tools to enhance the quality of our work. The usage was confined to two main areas:

Manuscript Polishing: LLMs were used to improve the language, clarity, and readability of the text. This included tasks such as correcting grammar and spelling, rephrasing sentences for better flow, and ensuring consistent terminology.

Code Generation Assistance: LLMs served as a supplementary tool in our development process. Their role was limited to generating boilerplate code, assisting with debugging, and providing syntax suggestions for implementing the experiments.

The core research ideas, experimental design, data analysis, and all conclusions presented in this paper were conceived and executed solely by the human authors. We have thoroughly reviewed all content, including any text or code initially suggested by an LLM, and take full responsibility for the accuracy and integrity of the entire manuscript.

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

# A APPENDIX

## A.1 LIMITATIONS AND FUTURE WORK

While our work establishes the importance of reward variance as a key evaluation dimension, we recognize several limitations that open promising avenues for future research.

- Scope of the Benchmark: Our current benchmark, RVB, is constructed based on the "help-fulness" subset of the Reward Model Benchmark (RMB). Future iterations should expand to include other critical alignment dimensions, such as "harmlessness" to provide a more holistic view of a reward model's teaching capabilities. Additionally, while our *Eval-Core* subset is meticulously curated for quality, expanding its size and diversity of tasks could further enhance the robustness of the findings.

- Generalizability of RLHF verification: Our RLHF experiments analyze short-horizon PPO-style (GRPO) training on a single helpfulness dataset (Anthropic/hh-rlhf) with one policy family (Qwen2.5-1.5B-Instruct), and the convergence AUC used in our regression is computed from reward curves that generally do not reach a clear performance plateau. These results should therefore be interpreted as preliminary evidence that RVB scores carry predictive signal for early-stage reward trajectories in our setting, rather than as a definitive characterization of long-run policy performance. A crucial next step is to repeat this analysis across a broader range of policy models, datasets, and optimization algorithms to test how robust these relationships are and whether different RLHF pipelines favor different variance profiles.

- Variance-aware training algorithms: Our RLHF verification deliberately fixes the optimization pipeline (GRPO with a Qwen2.5-1.5B policy on Anthropic/hh-rlhf) and varies only the reward model. We therefore do not include variance-regularized optimizers such as DGRO (Su et al., 2025), GVPO (Zhang et al., 2025), or GRPOVI (Yang et al., 2025b) as empirical baselines, because doing so would require retraining policies under each method for a comparable set of reward models, which is beyond our current compute budget. Instead, RVB is intended as a reusable evaluation layer: it can be directly applied to reward models or policies trained with these variance-aware algorithms, and a broader empirical study along this axis is a natural next step.

## A.2 SUPPLEMENTARY METRICS AND CORRELATION ANALYSIS

### A.2.1 $IQR(RSI)_{med}$: MID-RANGE SEPARATION

Definition (per prompt):

$$\text{IQR(RSI)}(p) = \frac{\text{IQR}(\{r_i\}_{i=1}^n)}{s_{\text{RM}}}, \qquad \text{IQR(RSI)}_{\text{med}} = \text{median}_p\big[\text{IQR(RSI)}(p)\big]. \qquad (9)$$

Here, $\{r_i\}_{i=1}^n$ is the set of n reward scores for the candidate responses to a given prompt $p$. The term $s_{RM}$ is the robust global scale factor defined in equation 1. The final metric is the median of IQR(RSI)(p) taken over all prompts.

**Semantics and Motivation**: IQR(RSI) measures how well an RM separates mid-quality responses (the middle 50%). It is resistant to tail outliers and works well with 5–10 candidates. Higher $IQR(RSI)_{med}$ indicates meaningful gaps among non-extreme responses, thereby providing richer gradient signals during broad exploration phases in policy training.

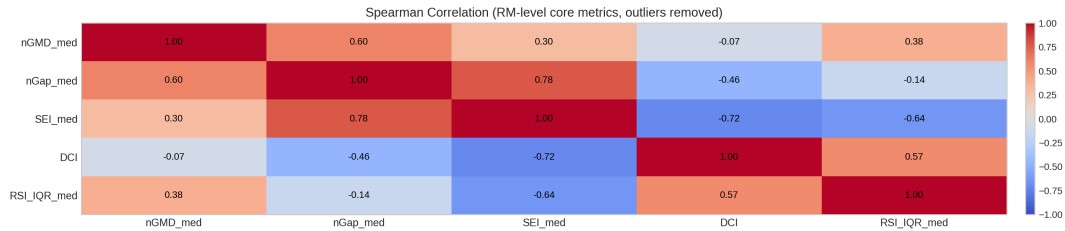

Figure 4: The Spearman Correlation of all five metrics.

| Feature | PC1 | PC2 | PC3 |
|---|---|---|---|
| RewardBench accuracy | 0.51 | 0.06 | 0.73 |
| $SEI_{med}$ | 0.63 | 0.32 | -0.01 |
| $nGMD_{med}$ | -0.28 | 0.94 | 0.01 |
| DCI | -0.52 | -0.07 | 0.69 |

Table 3: Principal component loadings for RewardBench accuracy and the RVB metrics across 23 RMs (features standardized to zero mean and unit variance). Signs are defined up to a global flip; we report one consistent orientation.

### A.2.2 $nGap_{med}$: NORMALIZED TOP-2 GAP

Definition (per prompt):

$$\text{nGap}(p) \;=\; \frac{r_{(1)} - r_{(2)}}{s_{\text{RM}}}, \qquad \text{nGap}_{\text{med}} \;=\; \text{median}_p\big[\text{nGap}(p)\big]. \tag{10}$$

where $r_{(1)} \geq r_{(2)}$ are the highest and second-highest scores for prompt $p$. $s_{RM}$ is the robust global scale factor defined in equation 1.

**Semantics and Motivation**: nGap captures decisiveness—whether the RM identifies a unique best response and assigns it a clear advantage. This metric is highly relevant to Best-of-N sampling. A high $nGap_{med}$ implies that the RM forms a steep reward peak around the optimal response, providing a clear and strong gradient direction that accelerates policy convergence.

### A.2.3 METRIC CORRELATION ANALYSIS

Figure 4 reports Spearman correlations among the five metrics at the RM level (outliers removed). Three takeaways emerge. First, the two "signal strength" measures are tightly coupled: SEI and nGap show a strong positive correlation ($\rho \approx 0.78$), while nGMD is only weakly to moderately correlated with them ($\rho \approx 0.30/0.60$), indicating that nGMD contributes complementary information beyond top-gap effects. Second, stability is largely decoupled from strength: DCI correlates negatively with SEI and nGap ($\rho \approx -0.72$ and $-0.46$) and is nearly orthogonal to nGMD ($\rho \approx -0.07$), confirming that decision-style stability captures a distinct dimension. Third, mid-range separation IQR(RSI) aligns with stability and opposes sharp concentration: it correlates positively with DCI ($\rho \approx 0.57$) and negatively with SEI ($\rho \approx -0.64$). These patterns motivate our primary set: **SEI** (concentration), **nGMD** (global separation), and **DCI** (cross-prompt stability).

### A.3 PCA OF ACCURACY AND RVB METRICS

### A.4 STABILITY UNDER TASK AND GENERATOR SHIFTS

### A.4.1 TASK-WISE NORMALIZATION

This subsection expands on the task-wise normalization analysis mentioned in Sec. 5.3. We recompute all metrics after normalizing scores within each task and examine how RM rankings change.

**Overall stability.** Rankings based on **nGMD** are highly stable under task-wise normalization (Spearman $\rho = 0.92$, mean $|\Delta\text{rank}| = 1.74$). **DCI** is also fairly stable ($\rho = 0.84$, mean $|\Delta\text{rank}| =$

2.52), while **SEI** is the most sensitive to task composition ($\rho = 0.50$, mean $|\Delta\text{rank}| = 4.26$). This aligns with the semantics: nGMD aggregates pairwise separation and is robust to scale shifts across tasks, whereas SEI depends on the entropy of per-task "score temperature," which varies more with task difficulty and candidate diversity.

**Leaders and invariances.** `URM-LLaMa-3.1-8B` remains the most consistent teacher: it keeps the top position on **DCI** (#1 $\rightarrow$ #1), stays strong on **nGMD** (#3 $\rightarrow$ #3), and slightly improves on **SEI** (#9 $\rightarrow$ #6), indicating that its decision style and separation capability are not artifacts of a particular task mix. Models with already strong separation—e.g., `Skywork-Reward-Gemma-2-27B`—are essentially unchanged on **nGMD** (#2 $\rightarrow$ #2).

**Where normalization matters.** Several models experience notable rank shifts once tasks are equalized:

- *SEI swings.* `beaver-7b-v2.0-reward` improves dramatically on **SEI** (#23 $\rightarrow$ #7; $\Delta = -16$) and `RM-Mistral-7B` also improves (#22 $\rightarrow$ #13; $\Delta = -9$), suggesting their confidence concentration was previously muted by specific task distributions. In contrast, `RAMO-Llama3.1-8B` (#7 $\rightarrow$ #21; $\Delta = +14$), `ArmoRM-Llama3-8B-v0.1` (#8 $\rightarrow$ #22; $\Delta = +14$), and `QRM-Llama3.1-8B-v2` (#6 $\rightarrow$ #19; $\Delta = +13$) drop markedly, indicating SEI was partly driven by task mix rather than uniform confidence across tasks.

- *nGMD is steady with localized adjustments.* Most models move little. The largest changes include `Skywork-Reward-V2-Qwen3-4B` improving (#14 $\rightarrow$ #8; $\Delta = -6$) and `Gemma-2B-rewardmodel-baseline` worsening (#10 $\rightarrow$ #17; $\Delta = +7$). These shifts are moderate compared to SEI.

- *DCI reorders a few mid-tier models.* `QRM-Gemma-2-27B` (#9 $\rightarrow$ #3; $\Delta = -6$) and `Gemma-2B-rewardmodel-baseline` (#13 $\rightarrow$ #6; $\Delta = -7$) gain substantially, while `tulu-v2.5-13b-uf-rm` falls (#3 $\rightarrow$ #11; $\Delta = +8$). Since DCI rewards cross-prompt stability of both separation and confidence, these changes suggest that some models' global consistency emerges more clearly after equalizing task scales, whereas others relied on task-specific strengths.

**Takeaway.** Task-wise normalization confirms that *separation* (nGMD) and *decision-style stability* (DCI) are intrinsically robust, while *confidence concentration* (SEI) can be heavily task-dependent. The fact that `URM-LLaMa-3.1-8B` preserves #1 on DCI and remains top-tier on nGMD under both settings supports its status as a reliable signal source for RLHF.

A.4.2 MULTI-GENERATOR ROBUSTNESS PROTOCOL

The task-wise normalization analysis above focuses on stability across tasks under a fixed candidate distribution. To make the dependence on the generator itself explicit and to facilitate future robustness checks, we first outline a generic protocol for instantiating RVB with an alternative generator and then report a small Qwen2-based case study.

**Prompt subset.** We start from the existing *Eval-Core* pool and draw a small stratified subset of prompts. Concretely, we sample $N_{\text{prompts}} \in [20, 30]$ prompts proportional to the number of prompts per task, using a fixed random seed for reproducibility.

**Alternative generator.** We select an open-source instruction-tuned model that is unrelated to GPT-4o, for example `Llama-3-8B-Instruct` or `Qwen2-7B-Instruct`. We keep the prompt formatting identical to the GPT-4o setting and generate $K = 9$ candidate responses per prompt with nucleus sampling (temperature 0.7, top-$p$ 0.95, `max_new_tokens` $= 512$). We discard exact duplicate strings and truncate to the first $K$ unique candidates per prompt.

**Re-scoring and RVB computation.** For each of the 23 reward models, we score all candidates from the alternative generator using the same inference and normalisation pipeline as in the main experiments. We then recompute SEI, nGMD, and DCI on this new candidate distribution, following Section 4, and aggregate them into a composite RVB score using the same weighting scheme as for the GPT-4o-based RVB variant. This yields an alternative RVB ranking of the 23 reward models conditioned on the new generator.

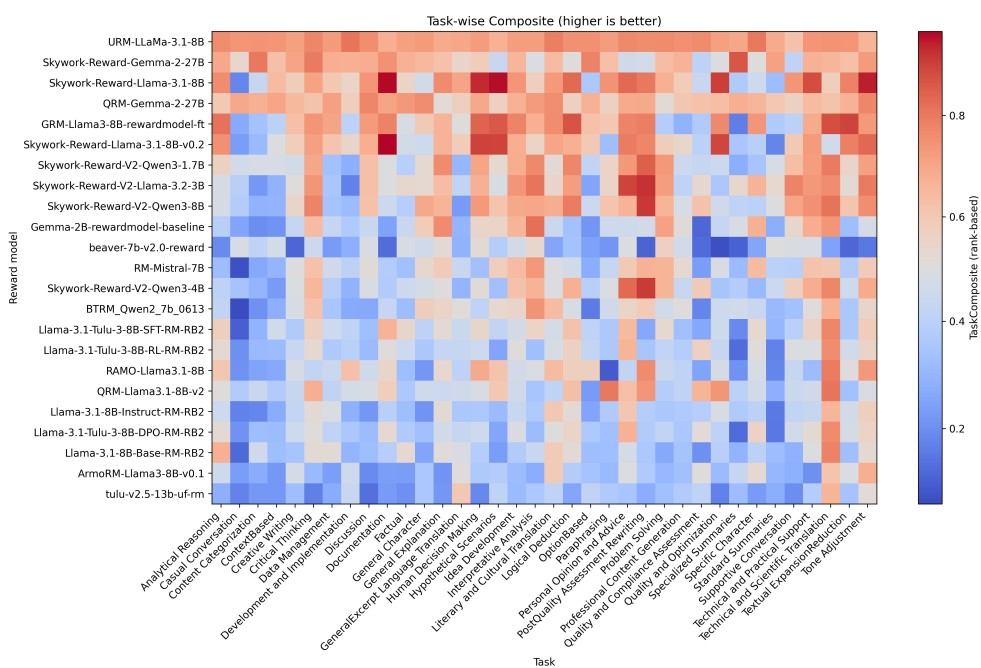

Figure 5: Heatmap of task-wise composite variance scores (rank-based across model–task pairs), where rows are models, columns are tasks, and warmer cells indicate stronger variance profiles. Models show distinct strengths on specific tasks.

**Comparing rankings across generators.** To quantify generator sensitivity, we compare the ranking induced by GPT-4o-based RVB with the ranking induced by the alternative-generator RVB. We compute Spearman's rank correlation coefficient $\rho$ between (i) the composite RVB scores across models, and (ii) each individual metric (SEI, nGMD, DCI) across models. Optionally, we bootstrap over prompts, recompute RVB scores on each bootstrap sample, and report a confidence interval for $\rho$.

**Preliminary Qwen2-based instantiation.** During the rebuttal period we instantiated this protocol with `Qwen2-7B-Instruct` as the generator. Starting from a stratified subset of 150 EvalCore prompts, we generated candidates with `Qwen2-7B-Instruct` and combined them with the original RMB responses. Using three representative reward models from different families, we computed an average percentile rank per candidate and enforced a low/medium/high quality structure by keeping only prompts for which we could select 9 candidates (three per quality tier). This filtering yields a `Qwen2-7B-Instruct`-based RVB instance with 10 prompts and 9 candidates per prompt. Running the full 23-RM evaluation on this pool, we obtain a Spearman correlation of $\rho \approx 0.64$ between the GPT-4o-based and `Qwen2-7B-Instruct`-based composite RVB rankings across models. Per-metric correlations are $\rho \approx -0.15$ (RSI_IQR_med), $0.49$ (nGMD_med), $0.68$ (nGap_med), $0.52$ (SEI_med), and $0.24$ (DCI), indicating moderate alignment with clear local rank swaps but preserved coarse tiers (top, middle, bottom). These preliminary results support our interpretation of RVB as providing a generator-conditional variance profile: changing the generator can move individual models within a tier, but does not completely reshuffle the leaderboard. A more systematic multi-generator study with larger prompt sets and multiple alternative generators is left for future work.

A.5 REGRESSION DETAILS FOR PPO REWARD AUC

To move beyond qualitative inspection of learning curves, we run an RM-level regression analysis to quantify how much variance information helps explain PPO training behaviour on a fixed RLHF

Table 4: **RM-level regressions to PPO reward AUC.** All models are fitted over the 23 reward models. Adding either RVB metrics or simple variance-style baselines to accuracy increases the explained variance in reward AUC from about one third to roughly one half, and combining all features does not further improve adjusted $R^2$ due to strong multicollinearity.

| Model | Predictors | Adjusted $R^2$ |
|---|---|---|
| (1) Accuracy only | RewardBench accuracy | 0.33 |
| (2) Accuracy + RVB | Accuracy + $\text{SEI}_{med}$ + $\text{nGMD}_{med}$ + DCI | 0.55 |
| (3) Accuracy + simple baselines | Accuracy + norm-std + entropy + Brier | 0.55 |
| (4) Accuracy + RVB + baselines | Accuracy + RVB metrics + baselines | 0.55 |

setup. For each of the 23 reward models, we run a short PPO run with shared hyperparameters on Anthropic/hh-rlhf (as in Sec. 6.1) and summarize the reward–step curve by the area under the raw reward trajectory (reward AUC). These PPO runs do not show clear monotonic convergence and rewards typically fluctuate in a narrow range (e.g., between $-1$ and $1$), but they still exhibit systematic differences in reward trajectories across reward models.

We then fit four linear regression models with reward AUC as the dependent variable and different combinations of predictors, all over the same 23 RMs: (1) accuracy only (RewardBench accuracy), (2) accuracy plus the three RVB metrics ($\text{SEI}_{med}$, $\text{nGMD}_{med}$, DCI), (3) accuracy plus three simple variance-style baselines (per-prompt normalized standard deviation, prediction entropy, and Brier score), and (4) a combined model using accuracy, the RVB metrics, and the baseline features. Table 4 reports the adjusted $R^2$ for these four specifications.

Accuracy alone explains about one third of the variance in reward AUC (adjusted $R^2 \approx 0.33$). Adding either the three RVB metrics or the three simple baselines increases the adjusted $R^2$ to $\approx 0.55$, and including both RVB and baseline features together does not improve the fit further, with the design matrix becoming nearly singular. This suggests that standard variance/entropy/Brier statistics and RVB capture largely the same linear structure with respect to PPO reward AUC, but RVB organizes this structure into explicit concentration (SEI), dispersion (nGMD), and stability (DCI) axes that are easier to interpret when reasoning about reward-model teaching signals. Given the small sample size and strong collinearity between predictors, we focus on this overall improvement in explained variance rather than on the significance or sign of individual coefficients.

### A.6 SEMANTIC ROBUSTNESS EVALUATION

#### A.6.1 EXPERIMENTAL OVERVIEW

We evaluate the semantic robustness of reward models by measuring the variance in reward scores when presented with semantically equivalent responses. This approach evaluates whether reward models maintain consistent scoring for responses that convey identical meaning through different expressions, which is crucial for stable preference learning in RLHF systems.

#### A.6.2 DATA PREPARATION AND SAMPLING STRATEGY

Our experimental dataset is derived from the RewardBench benchmark (Lambert et al., 2025), specifically utilizing the filtered subset of `allenai/reward-bench`. We sample all prompts from each of the four primary categories, totaling 800 samples for comprehensive analysis across diverse task types. For each sample, we extract the original prompt and the corresponding chosen response, which serves as our baseline for generating semantically equivalent variations.

#### A.6.3 SEMANTIC PARAPHRASE GENERATION FRAMEWORK

Our paraphrase generation utilizes GPT-4o (Hurst et al., 2024), guided by carefully crafted prompt templates, to produce semantically equivalent responses that maintain essential formatting and stylistic elements. We employ six distinct strategies that address different linguistic levels, ensuring a comprehensive range of paraphrasing possibilities. Prompt templates used for data generation are detailed in the Appendix A.6.7. These strategies include: Lexical substitution, focusing on synonym replacement while retaining core meaning; Sentence restructuring, reorganizing syntactic elements to preserve information; Natural rephrasing, using varied expressions for identical concepts;

Systematic alternative wording, ensuring semantic equivalence; Transformations into different expressions while preserving exact meaning; Comprehensive rewording with alternative phrasing that maintains original intent. Strict formatting preservation requirements are enforced to prevent bias during the rewriting process. These include maintaining the original tone (formal, informal, humorous, or serious), and preserving structural elements like markdown formatting, LaTeX expressions, and paragraph structures. The process uses a fixed temperature of 0.8 to balance randomness and coherence.

### A.6.4 QUALITY CONTROL AND FILTERING PIPELINE

Our quality assurance framework implements multiple layers of validation to ensure semantic equivalence and factual consistency. The primary filtering mechanism employs SimCSE (Gao et al., 2021) using the `sentence-transformers/all-MiniLM-L6-v2` model to compute semantic similarity scores between original and paraphrased responses. We establish a similarity threshold of 0.8, retaining only paraphrases that demonstrate high semantic alignment with their source material.

Factual consistency verification represents a critical component of our validation pipeline. We utilize Gemini-2.5-flash (Comanici et al., 2025) to conduct comprehensive factual alignment assessments, evaluating paraphrases on a continuous scale from 0 to 1. Our evaluation framework examines factual claims and assertions, numerical data and statistical information, temporal and nominal references, and logical relationships between concepts. Only paraphrases achieving consistency scores of 0.8 or higher proceed to the reward model evaluation phase.

The consistency scoring framework follows strict evaluation criteria where perfect alignment (1.0) requires exact factual matching without variations. High consistency (0.8-0.9) permits minor variations such as synonymous expressions that preserve meaning and implications. Moderate consistency (0.6-0.7) indicates some factual differences or omissions that slightly alter interpretation. Lower scores reflect increasingly significant factual discrepancies or contradictions that compromise the semantic equivalence requirement.

### A.6.5 SAMPLE SELECTION AND RANKING METHODOLOGY

Initially, we employed the complete *rewardbench* dataset for paraphrasing. However, following a detailed analysis, we refined our methodology by selecting a subset of paraphrased samples, thereby constructing an *Eval-Semantic* dataset with enhanced variance. To ensure comparability, we first applied **standardization** to the reward scores of each sample using the task-based global mean and standard deviation (Razin et al., 2025), bringing all scores onto a consistent scale. This normalization process enabled meaningful comparisons across samples. We then computed the variance of the standardized rewards within each sample. Since each sample may include multiple paraphrases, the reward models evaluated all valid paraphrases, producing standardized reward score lists for each sample.

For each model, the variance was calculated and used to rank the samples according to their reward stability. These rankings were then aggregated by summing the ranks from all models to obtain an overall ranking that reflects consensus across different reward architectures. Our model suite includes `Skywork/Skywork-Reward-Llama-3.1-8B`, `LxzGordon/URM-LLaMa-3.1-8B` and `Ray2333/GRM-llama3-8B-distill` for comprehensive coverage of different architectural approaches. Only samples with at least three valid paraphrases were considered to ensure sufficient statistical power for variance estimation.

Finally, for each task within *rewardbench*, we selected 200 samples, each containing approximately seven responses. The top 200 samples according to the aggregated ranking were selected, resulting in a high-quality evaluation set that maximizes the discriminative ability of semantic robustness evaluation by highlighting cases where reward models differ most in stability.

### A.6.6 ROBUSTNESS METRIC

The robustness metric is calculated based on task-specific normalization to quantify the consistency of a RM in evaluating semantically equivalent responses within each task category (e.g., Chat, Reasoning, Safety). For each response, the normalized reward score is given by:

$$z_i = \frac{r_i - \mu}{\sigma} \tag{11}$$

where $r_i$ is the raw reward score for the $i$-th sample, and $\mu$ and $\sigma$ are the global mean and standard deviation of all reward scores, respectively.

The mean of the normalized scores, $\mu_z$, is computed as:

$$\mu_z = \frac{1}{n} \sum_{i=1}^{n} z_i \tag{12}$$

Finally, the robustness metric is given by:

$$\text{Robustness Metric} = \frac{1}{n} \sum_{i=1}^{n} (z_i - \mu_z)^2 \tag{13}$$

This metric quantifies the variability in the normalized reward scores, providing a measure of how consistently the reward model evaluates semantic similarity across different responses.

### A.6.7 COMPREHENSIVE EXPERIMENTAL RESULTS

We present comprehensive semantic robustness evaluation results across eight reward models spanning different architectural approaches and training methodologies. Table 5 summarizes the performance across four task categories, reporting both mean semantic variance and standard deviation for each model-category combination.

Table 5: Semantic robustness evaluation results across reward models and task categories. Values represent mean semantic variance. Bold entries highlight models used in our primary analysis.

| Reward Model | Chat | Chat Hard | Reasoning | Safety | Overall | RewardBench Score |
|---|---|---|---|---|---|---|
| **Skywork-Reward-V2-Qwen3-8B** | 0.155 | 0.187 | 0.266 | 0.159 | 0.192 | N/A |
| **Skywork-Reward-Llama-3.1-8B** | 0.132 | 0.264 | 0.110 | 0.108 | 0.154 | 92.5 |
| Eurus-RM-7b | 0.130 | 0.093 | 0.234 | 0.124 | 0.145 | 82.8 |
| Skywork-Reward-V2-Llama-3.1-8B | 0.122 | 0.148 | 0.216 | 0.070 | 0.139 | N/A |
| **URM-LLaMa-3.1-8B** | 0.125 | 0.174 | 0.136 | 0.047 | 0.121 | 92.9 |
| GRM-gemma2-2B-rewardmodel-ft | 0.087 | 0.144 | 0.097 | 0.078 | 0.101 | 88.4 |
| **GRM-llama3-8B-distill** | 0.058 | 0.108 | 0.167 | 0.055 | 0.097 | 86.2 |
| RM-Mistral-7B | 0.034 | 0.059 | 0.077 | 0.025 | 0.049 | 80.4 |

The results reveal substantial heterogeneity in semantic robustness across different model architectures and task domains. Models demonstrate varying sensitivity to semantic variations, with Chat Hard tasks often exhibiting the highest variance across most models, suggesting increased difficulty in maintaining consistent preferences for complex conversational scenarios.

Reasoning tasks show mixed patterns: some models (e.g., Skywork-Reward-V2-Qwen3-8B) exhibit particularly high variance (0.266), while others (e.g., Skywork-Reward-Llama-3.1-8B) maintain relatively stable performance (0.110).

Safety-related tasks generally demonstrate lower semantic variance across most models, potentially reflecting more standardized training approaches for safety-critical scenarios. Notably, the URM-LLaMa-3.1-8B model shows exceptionally low variance in safety tasks (0.047), suggesting robust consistency in safety-related preference judgments.

Overall performance rankings reveal that RM-Mistral-7B achieves the lowest overall semantic variance (0.049), indicating superior robustness to semantic variations. In contrast, Skywork-Reward-V2-Qwen3-8B exhibits the highest overall variance (0.192), suggesting potential sensitivity to linguistic variations that may impact deployment stability.

These comprehensive findings suggest that semantic robustness varies significantly across model architectures and training approaches, with potential implications for RLHF system stability.

You are producing three answers of deliberately different quality tiers (LOW, MEDIUM, HIGH) for the *same* user prompt to support a Response Variance benchmark.

Language rule:
- Write in the same language as the user prompt. If unclear, default to English.

Quality definitions (do not disclose to the user):
- HIGH: Fully correct, complete, well-structured. Follows all constraints. Clear reasoning when helpful. No hallucinations.
- MEDIUM: Generally correct but with minor omissions, shallow reasoning, weaker structure, or small style issues.
- LOW: Safe but flawed. Stay on-topic yet intentionally underperform by one or more of: brief/vague, missing key constraints, shallow or partially incorrect reasoning. Do NOT produce unsafe/offensive content. Do NOT refuse the task.

Stylistic/structural diversity:
- Vary structure, tone, and detail across tiers (list vs. prose; concise vs. detailed).
- Do NOT mention the existence of tiers.
- Rough length guide (flexible for English): LOW 60–120 tokens, MEDIUM 120–200, HIGH 180–300.

Output format:
Return ONLY a JSON object:
{{
  "prompt": "<the prompt text you answered>",
  "answers": [
    {{"quality_tier": "low", "answer": "<text>"}},
    {{"quality_tier": "medium", "answer": "<text>"}},
    {{"quality_tier": "high", "answer": "<text>"}}
  ]
}}

User prompt to answer:
{prompt}

Figure 6: The Data Generation Prompt for *Eval-Core*.

Task Content:
I aim to rework the content into a rendition that retains a comparable semantic meaning, with the additional requirement of preserving certain elements of its original stylistic features.

CRITICAL REQUIREMENTS:
Maintain exact tone and style (formal/informal/humorous/serious)
Preserve all markdown formatting (headers, bold, italic, links, code blocks)
Keep all list structures intact (bullet points, numbered lists, nested lists)
Maintain all LaTeX mathematical expressions unchanged
Preserve paragraph structure and organization
Keep similar length and complexity level
Maintain code syntax highlighting and structure
Preserve special formatting (tables, quotes, etc.)

Rewrite Prompt Templates
Use these templates for content variation while preserving format:
Synonym Replacement: "Please rewrite using synonyms and alternative expressions while maintaining exact meaning and all formatting requirements above."
Sentence Restructuring: "Please restructure the sentences while preserving all original meaning and formatting requirements above."
Expression Rephrasing: "Please rephrase using different expressions while keeping identical meaning and all formatting requirements above."
Semantic Equivalence: "Please create a semantically equivalent version using alternative wording while preserving all formatting requirements above."
Expression Transformation: "Please transform into a different expression while maintaining exact same meaning and all formatting requirements above."
Phrasing Alternatives: "Please reword using different phrasing while preserving all original meaning and formatting requirements above."

User prompt to answer: {prompt}

Figure 7: The Data Generation Prompt for *Eval-Semantic*.

## A.7 PROMPT TEMPLATE DETAILS

You are a strict factual consistency expert. Your task is to meticulously compare the original text and the paraphrased version for any discrepancies in facts, even subtle ones. Be critical: do not assume consistency unless every detail matches perfectly.

First, list all key factual elements from the original text (e.g., claims, numbers, dates, names, events, specific details).
Second, list all key factual elements from the paraphrased text.
Third, compare them element-by-element, noting any additions, omissions, alterations, or nuances that could change meaning (e.g., a word change implying different intent).
Finally, rate the factual consistency on a scale of 0-1 based on the comparison.

Scoring guidelines (be strict - deduct points for any mismatch):
- 1.0 = Completely factually consistent (every fact, claim, and detail matches exactly, no variations at all)
- 0.8-0.9 = Mostly consistent (very minor variations like synonyms that don't alter meaning or implications)
- 0.6-0.7 = Partially consistent (some factual differences, omissions, or additions that slightly change interpretation)
- 0.4-0.5 = Inconsistent (significant factual differences, alterations, or contradictions)
- 0.0-0.3 = Completely inconsistent (major contradictions or entirely different facts)

Examples:
Example 1:
Original: "The event occurred on July 4, 2023, in New York, attended by 500 people."
Paraphrased: "The gathering happened on Independence Day 2023 in NYC, with about 500 attendees."
1. Consistency score (0-1): 0.9
2. Explanation: Dates match (July 4 is Independence Day), locations match (NYC is New York), numbers are exact. Minor variations in wording (event/gathering, people/attendees) don't change meaning, but 'about' introduces slight approximation.
3. Any factual differences found: Slight approximation in attendance ('about 500' vs. exact '500').

Example 2:
Original: "Apple released the iPhone 15 in September 2023, featuring a 48MP camera."
Paraphrased: "In 2023, Apple launched their new phone with a high-resolution camera."
1. Consistency score (0-1): 0.6
2. Explanation: Month is omitted (September missing), model name changed vaguely ('iPhone 15' to 'new phone'), camera spec generalized ('48MP' to 'high-resolution'). These omissions and generalizations alter specific details.
3. Any factual differences found: Omission of exact month, vague model name, loss of precise camera specs.

Original text:
{original}

Paraphrased text:
{paraphrase}

Respond exactly with:
1. Consistency score (0-1):
2. Explanation:
3. Any factual differences found:

After your response, self-criticize: Did I miss any subtle differences? If yes, adjust the score.

Figure 8: The Fact Consistency Check Prompt of *Eval-Semantic*.

