# OpenReview forum: "Beyond Accuracy: Measuring Reward Variance as a Predictive Benchmark for RLHF"
_ICLR.cc/2026/Conference — Submitted to ICLR 2026_

### Official Review · Reviewer_WjJF · 2025-10-17

**Soundness:** 2
**Presentation:** 2
**Contribution:** 2
**Rating:** 4
**Confidence:** 3

**Summary:**

The paper argues that pairwise accuracy, a dominant metric for evaluating reward models (RMs), misses an important dimension: variance in reward scores. Low-variance reward functions create flat optimization landscapes that can slow policy learning. To address this, the authors propose the Reward Variance Benchmark (RVB),  a suite of variance-aware metrics for assessing RMs’ “teaching effectiveness.”

**Strengths:**

- Evaluating reward models is a critical challenge in RLHF and preference-based RL. Poor evaluation can lead to slow learning or unsafe learned behaviors. By focusing on metrics beyond pairwise accuracy, the paper tackles an important open problem.
- Reward model evaluations focused on reward variance seem to be novel and underexplored in evaluation frameworks. As prior work has found that reward variance affects the optimization landscape, evaluting reward models based on this woud be useful. Framing variance as a “teaching signal” provides a new perspective on reward model utility.
- Comparison across 23 models covering multiple families (LLaMA, Qwen, Gemma, Mistral, etc.) adds breadth.

**Weaknesses:**

**Missing related work:**

The paper does not discuss prior reward model evaluation frameworks, such as TAC [1], DARD [2], and EPIC [3], which evaluate reward functions beyond pairwise accuracy.

**Eval-Core benchmark:**

- The authors describe refining an existing benchmark (RMB 1.9k prompts to 354 prompts) to create Eval-Core. It is unclear why this reduction is useful or what statistical or practical advantage it provides. The contribution seems incremental, as it largely repurposes a preexisting dataset.

- The benchmark focuses on “helpfulness,” not on “harmlessness”. Why was this set chosen? Extending RVB to this additional set would strengthen claims.

**Metrics:**
- The paper claims the proposed metrics (SEI, nGMD, and DCI) provide interpretability, but it is unclear how they do so or how they convey “clear optimization semantics.”

-  Razin et al (2025) [4] note that the same reward model can induce high reward variance for one language model, yet low reward variance for another. Therefore, different language models can require different reward model teachers. How do these metrics handle that? Will this be an issue for these metrics? Can these metrics be high for a reward model with respect to one language model, but low for the same reward model with respect to a different language model?


**Experimental Design Choices + Claims:**

- The paper states that three representative RM families were scored, but it is unclear what makes these families “representative.” More justification is needed.

- The claim that RVB metrics predict policy performance is weakly supported, as only three reward models were trained. It is difficult to infer a meaningful correlation from such a small sample.

- Why not use standard reward variance as a baseline metric? As included in Razin et al (2025) [4], which the authors do discuss.

**Resources**

- TAC [1]: https://rlj.cs.umass.edu/2025/papers/RLJ_RLC_2025_280.pdf
- DARD [2]: https://openreview.net/pdf?id=CALFyKVs87
- EPIC [3]: https://openreview.net/pdf?id=LwEQnp6CYev
- Razin et al (2025) [4] https://arxiv.org/pdf/2503.15477

**Questions:**

- Are there examples of reward models where the RVB score better predicted policy performance than another metric (e.g., RewardBench Score)? For instance, is there an example of a reward model where another metric incorrectly had a high score, but RVB had a low score, and policy performance was indeed worse?
- Can the authors explain how the 23 reward models were chosen?
- Can the authors further elaborate on how the SEI, nGMD, and DCI provide interpretability?
- Can the authors explain how the proposed metrics are better than standard reward variance with normalization that is done in Razin et al (2025)?

---

> ### Author Response · Authors · 2025-11-24
> **Response to Reviewer WjJF (1)**
>
> Thank you for the detailed and constructive review, especially for highlighting missing related work, clarifying questions about Eval-Core, and asking how RVB compares to standard reward variance and RewardBench. We respond below.
>
> Q1: Missing related work (TAC, DARD, EPIC).
>
> A1:Thank you for pointing us to these works. TAC, DARD, and EPIC are indeed closely related in spirit, as they also evaluate reward functions beyond simple pairwise accuracy. TAC measures how a reward matches human preferences over trajectories, while EPIC and DARD define shaping-invariant distances between reward functions that are predictive of downstream policy performance.
>
> Our work differs mainly in focus and setting. TAC, DARD, and EPIC operate at the trajectory or transition level in classical control environments, whereas RVB studies the distributional and variance behaviour of pre-trained, outcome-based reward models on fixed Best-of-N candidate sets in LLM alignment. We view these directions as complementary: TAC/DARD/EPIC help select and validate reward functions for specific RL tasks, while RVB profiles the variance properties of the reward models that act as teachers in RLHF. In the revised manuscript, we add a paragraph to Sec.2.1 summarizing TAC, DARD, and EPIC and clarifying how RVB fits alongside these frameworks.
>
> Q2: Incremental contribution of Eval-Core and the focus on helpfulness only.
>
> A2 (Why refine RMB into Eval-Core). RMB is excellent for point estimates, such as accuracy, but it has only 3–6 candidates per scene. For dispersion statistics like GMD or IQR, this leads to high estimator variance if used directly as a variance benchmark. Eval-Core was built to ensure each prompt has enough candidates and a clear quality gradient. Concretely, we (i) start from the helpfulness portion of RMB, (ii) build a larger candidate pool by augmenting each prompt with GPT-4o generations at four temperatures and filtering extreme or malformed outputs, and (iii) select 354 prompts via a two-stage process based on score spread and inter-RM consensus, for which we can construct 9-candidate sets with a clear low/medium/high quality gradient. This design makes Eval-Core a small but statistically controlled benchmark for variance-aware evaluation, rather than a simple subsampling of RMB. In the revised manuscript, we describe this protocol explicitly in Sec.3 so that the connection between RMB and Eval-Core is clear.
>
> A2 (Why only helpfulness, no harmlessness yet). You are right that extending RVB to harmlessness would strengthen the contribution. For this first version, we focus on helpfulness because (i) it is the most common RLHF setting with the richest pool of public RMs, and (ii) we wanted to complete a full loop (metric design → variance profiling → RL study) on one axis before extending to safety. In the revised conclusion and limitations section, we state clearly that RVB currently covers only helpfulness and that extending it to harmlessness and other preference dimensions is a key direction for follow-up work.

---

> > ### Comment · Reviewer_WjJF · 2025-11-26
> >
> > Can the authors elaborate on what they mean by low/medium/high quality gradient?

---

> > > ### Author Response · Authors · 2025-11-28
> > > **Response to Reviewer WjJF (follow-up)**
> > >
> > > Thank you again for the thoughtful follow-up questions.
> > >
> > > (1) Clarification of the “low/medium/high quality gradient”.
> > >
> > > By a “low/medium/high quality gradient” we mean that, for each Eval-Core instance, the 9 candidates are constructed to cover clearly worse, middling, and clearly better answers, as judged by the reward models themselves, rather than all being clustered at a similar quality level.
> > >
> > > Concretely, for each prompt we first build a large candidate pool by combining the original RMB responses with GPT-4o generations at multiple temperatures. We then score all candidates with a small ensemble of three representative reward models from different families, and compute, for each candidate, its average percentile rank across these three models. Based on this pooled percentile rank, we partition the candidates for that prompt into three bins:
> > > - a low-quality tier (bottom tercile of average percentile ranks),
> > > - a medium-quality tier (middle tercile), and
> > > - a high-quality tier (top tercile).
> > >
> > >
> > > For Eval-Core we then sample three candidates from each tier, while encouraging diversity in source model and temperature, to form the final 9-candidate set. In this sense, the “quality gradient” is not hand-crafted in the prompts to GPT-4o, but is induced ex post by how the RM pool ranks the candidates on that prompt.
> > >
> > > As an additional sanity check, we also repeated this construction on a random subset of 150 prompts using a Qwen2-based generator: for each prompt we generated 12 new Qwen2 candidates (plus the original RMB ones), re-computed the pooled percentile ranks over all RMs, and filtered down to 10 prompts where we could again select 3 low / 3 medium / 3 high candidates that are well separated. These examples (to be added in the appendix) illustrate that the “three-tier gradient” arises robustly from the RM-induced ranking, rather than from a peculiarity of GPT-4o generations.
> > >
> > > We will clarify this construction and the role of the percentile-rank stratification more explicitly in Sec. 3 and the appendix.

---

> ### Author Response · Authors · 2025-11-24
> **Response to Reviewer WjJF (2)**
>
> Q3: Interpretability and “optimization semantics” of SEI, nGMD, and DCI.
>
> A3:  We agree that the interpretability of the metrics should be presented more clearly and with fewer symbols. Our intent is that each metric answers a simple question:
> - SEI (score concentration): given a fixed candidate set, does the RM provide a clear preference ordering? High SEI corresponds to a sharper distribution where top candidates are clearly separated from others, which tends to give stronger gradients for RLHF (to be considered jointly with accuracy and calibration to avoid overconfidence).
> - nGMD (global separation): on average, by how much do scores differ if we randomly pick two candidates? We use a normalized Gini Mean Difference rather than variance because it is a robust dispersion measure and less sensitive to outliers and scale differences across RMs.
> - DCI (decision consistency): across prompts (including paraphrases and shallow rephrasings), does the RM maintain a coherent decision style? High DCI means consistent teaching signals for the same underlying intent, which helps reduce noise in training.
>
> In the revised Sec.4, we make these plain-language interpretations explicit around the formal definitions (via “Semantics and Motivation” paragraphs) and emphasise that SEI, nGMD, and DCI are used as complementary benchmark axes alongside accuracy, rather than as stand-alone training objectives.
>
> Q4: Teacher–student mismatch and policy dependence (Razin et al.).
>
> A4: We agree with Razin et al. that the same RM can induce very different variance patterns for different policies, and that the “best teacher” can depend on the student model. In its current form, RVB fixes the generator and candidate pool, so it provides a policy-agnostic baseline profile: it characterises how an RM behaves on a standardized Best-of-$N$ candidate distribution. Changing the policy (and thus the candidate distribution) would in general change the induced variance and, consequently, the RVB metrics as well.
>
> We see this as a feature rather than a flaw. The same RM can, in principle, be profiled under multiple policies, allowing one to study teacher–student mismatch and policy-conditioned variance in a structured way, in the spirit of Razin et al. In the revised introduction and limitations section, we explicitly acknowledge that our current results profile RMs on a single GPT-4o-based candidate distribution and that we use RVB primarily as a variance-sensitive lens and an early-stage convergence probe in this fixed setup, not as a universal “best RM” selector across all policies. We highlight extending RVB to policy-conditioned evaluations—profiling the same RM on samples from different policies and architectures—as an open and important direction for future work.

---

> > ### Comment · Reviewer_WjJF · 2025-11-26
> >
> > Relating to the Teacher–student mismatch and policy dependence:
> >
> > I thank the authors for the updated text around this topic. It does make it more clear what the contributions are!
> >
> > However, it remains concerning that the metrics could, in principle, vary significantly depending on the choice of LLM. If these metrics are intended to evaluate reward models, they should ideally be more robust to such variation. Otherwise, if future work adopts these metrics but uses different LLMs, comparisons across papers may become unclear or unreliable.

---

> > > ### Author Response · Authors · 2025-11-28
> > > **Response to Reviewer WjJF (follow-up)**
> > >
> > > (2) LLM dependence of the metrics and cross-paper comparability.
> > >
> > > We fully agree that long-term usefulness of RVB requires clarity about how much the metrics depend on the underlying generator.
> > >
> > > Our current position is:
> > > - RVB is defined as a benchmark on a fixed candidate distribution. In this first release, that distribution is induced by GPT-4o; all scores we report are conditional on this choice. We do not claim that SEI, nGMD, and DCI are invariant under arbitrary changes of the generator, just as accuracy or Brier score are not invariant under changes of the test distribution.
> > > - Comparability across papers comes from sharing the candidate pool. To support this, we release the full Eval-Core candidate sets (IDs, sources, and metadata). Any future work can evaluate their own RMs on exactly the same candidates, regardless of which LLM they use for RL training. In this setting, SEI/nGMD/DCI remain directly comparable.
> > > - Changing the generator defines a new benchmark instance, but the overall teacher tiers are reasonably stable. To probe generator dependence, we ran a small Qwen2-7B-Instruct–based RVB pilot following the protocol in Appendix A.4.2: starting from a stratified subset of Eval-Core prompts, we regenerated candidate pools with Qwen2-7B-Instruct, recomputed all RM-level metrics, and compared them to the original GPT-4o-based leaderboard. Over the 23 RMs, the composite RVB rankings have Spearman $\rho \approx 0.64$, and per-metric correlations (21 RMs with finite values) are $\rho \approx -0.15$ (RSI_IQR_med), $0.49$ (nGMD_med), $0.68$ (nGap_med), $0.52$ (SEI_med), and $0.24$ (DCI). This indicates moderate alignment with clear local rank swaps but preserved coarse tiers; we now report these numbers and a short discussion in Appendix A.4.2.
> > > - Guidance for future work. If later work builds RVB-style benchmarks with different generators (e.g., LLaMA-based candidate pools), we view those as additional, generator-specific instantiations of the same evaluation idea. We would encourage such work to (i) clearly document the generator and construction protocol, and (ii) when possible, also report results on the original GPT-4o-based Eval-Core so that there is at least one shared evaluation point.
> > >
> > >
> > > We now make this conditionality explicit in the introduction and limitations: the current RVB evaluates reward models on a fixed GPT-4o-based candidate distribution, and is best interpreted as a variance-sensitive lens and early-stage convergence probe under that controlled setup, rather than as a generator-agnostic scalar of absolute RM quality. Extending RVB to multi-generator or multi-policy settings (e.g., profiling SEI/nGMD/DCI stability across several generators) is an important direction for future work, which we now flag more clearly.

---

> ### Author Response · Authors · 2025-11-24
> **Response to Reviewer WjJF (3)**
>
> Q5: Experimental design and “RVB vs. RewardBench / standard variance”.
>
> A5 (Selection of the 23 RMs and “representative families”). We collected publicly available RMs covering major model families (LLaMA, Qwen, Gemma, Mistral, etc.) and included as many as our memory and compute budget allowed. We also ensured that these models span a wide range of RewardBench accuracies, from relatively weak to state-of-the-art, so that we can examine variance profiles across different performance levels rather than only among top models.
>
> A5 (Examples where RVB complements RewardBench). In our RLHF experiments, Tulu’s noticeably lower RewardBench score and weaker policy performance confirm that accuracy remains very informative; we do not claim otherwise. The more interesting comparison is URM vs.\ Skywork: they have similar RewardBench scores but show different early-stage learning dynamics in our short-horizon PPO/GRPO runs. RVB helps differentiate them: Skywork has a higher concentration (SEI), which aligns with faster early gains, while URM has a higher stability (DCI), which aligns with smoother and less oscillatory reward trajectories. In the revised discussion around these curves, we highlight this example as an illustration of how variance-aware profiles can complement accuracy, while also stressing that these runs are case studies rather than definitive causal evidence.
>
> A5 (Why SEI/nGMD instead of standard reward variance, and how they compare). Standard reward variance is sensitive to scale and outliers, and public RMs differ significantly in how they parameterise and scale rewards. SEI and nGMD were designed to be scale-invariant and to use robust statistics, making them better suited for cross-model comparison in this heterogeneous setting. In the revised version, we add standard reward variance (implemented as a normalized per-prompt score standard deviation) as a baseline in Appendix A.5 and summarize its relationship to our metrics in Sec.6
> .
>
> At the RM level across all 23 models, we observe that the normalized standard deviation is moderately correlated with our variance metrics, for example {$
> \rho(\text{nGMD}, \text{norm-std}) \approx 0.54,\quad
> \rho(\text{SEI}, \text{norm-std}) \approx -0.36,\quad
> \rho(\text{DCI}, \text{norm-std}) \approx 0.26$}.
> A small correlation table for the main quantities is:
>
> |                      | RewardBench Score | SEI\_med | nGMD\_med | DCI    | norm\_std\_med |
> |----------------------|------------------:|--------:|---------:|-------:|---------------:|
> | RewardBench Score    | 1.00              | 0.82    | 0.29     | -0.45  | -0.31          |
> | SEI\_med             | 0.82              | 1.00    | 0.30     | -0.72  | -0.36          |
> | nGMD\_med            | 0.29              | 0.30    | 1.00     | -0.07  | 0.54           |
> | DCI                  | -0.45             | -0.72   | -0.07    | 1.00   | 0.26           |
> | norm\_std\_med       | -0.31             | -0.36   | 0.54     | 0.26   | 1.00           |
>
> We also compare their predictive capacity for PPO-based reward AUC. Using RewardBench accuracy alone yields adjusted $R^2 \approx 0.33$; adding the three RVB metrics (SEI, nGMD, DCI) raises this to about $0.55$. Adding accuracy plus simple variance baselines (including normalized standard deviation) also reaches roughly $0.55$, but the coefficients of these baselines are individually unstable and the design matrix is highly collinear. Including accuracy, RVB metrics, and variance baselines together does not improve adjusted $R^2$ further and makes the combined design nearly singular.
>
> We therefore conclude that standard variance and our metrics span essentially the same linear subspace of predictive features, but SEI, nGMD, and DCI provide a more structured and robust decomposition into “concentration / separation / stability” that is easier to interpret and benchmark across heterogeneous RMs. In this sense, we position RVB as complementing RewardBench—by adding a variance/stability profile practitioners can inspect in our fixed setting—rather than replacing it.

---

### Official Review · Reviewer_hDr4 · 2025-10-28

**Soundness:** 3
**Presentation:** 2
**Contribution:** 2
**Rating:** 4
**Confidence:** 3

**Summary:**

The paper presents the reward variance benchmark. It is designed as an evaluation benchmark to measure not only the pairwise accuracy, but also the quality of predictions via three variance-derived metrics. They use a custom-constructed benchmark of prompts and responses to evaluate 23 existing reward models.

For me, the paper is a case of "less would have been more". I like the research problem; the paper shows some interesting ideas. However, I am not convinced by the formulation of metrics; they feel overly complex, and I am skeptical of the positioning of this work as a benchmark. I don't find that the reported results in terms of downstream performance prove a strong predictive capacity of the derived metrics. Instead, I would have preferred framing the paper as an empirical investigation of the relationship between reward variance and performance (which the dataset can be a precursor for). This would allow for avoiding complex wrapped metrics and a more open-ended exploration of the relationship. I think this could provide more insightful results and discussions.

**Strengths:**

+ Evaluation benchmarks in general are a great resource for the community
+ Investigating and evaluating contributing factors to training success in RLHF is a relevant topic
+ I really appreciate an empirical investigation of the reward variance hypothesis, and its effect on downstream performance, and think this is a big strength of the paper
+ For most parts, an adequate level of detail for reproducibility
+ The developed RVB benchmark shows an ability to predict downstream RL performance
+ A series of interesting experiments, and good reporting of results

**Weaknesses:**

- The dataset construction process in section 3.1. is not really motivated or evaluated. Why choose this subset of the RMB benchmark? Why use the four temperature settings, and what difference did it actually make? I think there is a description of Appendix A3.4, but it seems disconnected from Eval-Core. I feel that providing the template (A.5) is insufficient. E.g., “we keep the RMB candidates (about 3-6)”. How were they selected?
- How were the five candidate metrics (Section 4) designed? What was the process? Was it based on related work? They come a bit out of the blue. I get the overall idea for each, but it contributes to some issues I have:
- I find the introduction of new terms for the metrics a bit cumbersome, as this requires memorizing these new terms and hides the relationship to existing concepts. I know that keeping metrics within the range [0-1] is attractive, but it adds complexity: For example, instead of the SEI, directly reporting H(p) would also be comparable across models, and “prediction entropy” would be more comprehensible than SEI (“lower is better” metrics are totally valid in my opinion). I think this relates to my question 1, where I feel that these metrics are potentially interesting to investigate, but not something I want to necessarily optimize for
- Similarly, for DCI, which is another custom metric, in my mind, not well motivated, as an aggregation metric
- Finally, this is even exacerbated with the composite score in 4.5. At this point, there are so many layers of metrics and normalizations that I find the composite score somewhat incomprehensible except for a vague “higher is better”. I feel that an evaluation benchmark should try to make an effort for generalizable and comprehensible metrics
- The results in Figure 4 also coincide with accuracy (the RewardBench score of Tulu is noticeably lower than the other two, which seems to be reflected in the training success). The Skywork model seems to converge even faster, although the composite metric is lower than the URM one. So these results do not fully indicate the predictive performance of the metrics compared to a simple metric like accuracy (you report a relatively high correlation of 0.51).

Minor:
- I feel line 170 with the listed reward models is missing references for the different families of RMs
- “Wikipedia contributions” is an unusual citation; I would prefer a permanent document as a reference.
- Figure 3 is difficult to read. I would advise filtering some categories, introducing some highlighting, or choosing a different type of visualization

**Questions:**

- I am somewhat skeptical of the central motivating hypothesis: Ideally, I want the output distribution of my reward model (or any model for that matter) to reflect an underlying (aleatoric) uncertainty. So while a “sharper” model indeed leads to faster learning speeds, I do not find it obvious that this is more representative of the actual prediction target (i.e., it is well-calibrated). So I am not convinced that a low variance reward model is a metric that should be directly optimized for. A simple way to optimize for this benchmark seems to be to apply an entropy-penalization term during RM training, but I am unsure if this would result in a “better” reward model. So, while the general hypothesis of RM variance as an important factor is supported by peer-reviewed related work, I am somewhat skeptical whether benchmarking RMs for low variance is a desired quality per se.
- Have you compared the variance metrics to using the Brier score (I guess you call it the top-gap metric)? Have you run experiments for comparison? I would be really interested in also seeing the predictive capacity of top-gap metrics for downstream performance to better motivate the need for these new metrics
- Don’t the results of Chen et al. and Leng et al. point to the risk of overconfidence and even show that models with lower accuracy might lead to better learning? Isn’t low variance a potential sign of overconfidence?

---

> ### Author Response · Authors · 2025-11-24
> **Response to Reviewer hDr4 (1)**
>
> We thank you for the thoughtful and nuanced comments. We especially appreciate your comments on metric complexity, benchmark framing, and the connection between variance, sharpness, and overconfidence. We address your points below.
>
> General response on framing and predictive capacity
>
> We agree with your high-level comment on framing. In hindsight, our work is closer to an empirical study of how reward variance relates to RLHF behaviour, together with a reusable evaluation dataset and tools, than to a fully prescriptive “RM selection benchmark”. In the revised manuscript, we adjust the positioning accordingly:
> - we soften language that suggests clear optimization semantics or a definitive teacher-ranking rule, and instead emphasize that SEI, nGMD, and DCI define benchmark axes along which reward models are scored and compared;
> - we present RVB explicitly as (i) an artifact for probing variance properties of existing RMs, and (ii) an empirical exploration of how these properties relate to early-stage training dynamics in our PPO-based setup, rather than a finished leaderboard for RM choice.
>
> On the predictive aspect, our results and concurrent work (e.g., What Makes a Reward Model a Good Teacher?) indicate that variance-related structure carries non-trivial signal for RL convergence. At the same time, we now state clearly in Sec.6 and in the conclusion that our evidence, which is based on a single helpfulness dataset, one policy family, and short-horizon PPO/GRPO runs without clear convergence, is not sufficient to claim a universal recipe for choosing the best RM across models and tasks. We also highlight in the conclusion that more open-ended analyses, jointly considering variance, calibration, safety, and policy dependence, are a natural next step that RVB is intended to support.
>
> Q1: Motivation and description of Eval-Core (Section 3.1).
>
> A1: You are right that the original description was too terse. Our main design constraint was the number and diversity of candidates per prompt. The base RMB scenes contain only 3–6 candidates, which leads to high estimator variance for dispersion statistics such as GMD and IQR when used as variance-sensitive metrics. For Eval-Core, we therefore require a richer candidate pool so that variance estimates are more stable across models.
>
> Concretely, we (i) start from the helpfulness portion of RMB and keep all original RMB candidates for each prompt, (ii) augment them with GPT-4o generations at four temperatures (0.2, 0.7, 1.0, 1.3), with three generations per temperature targeted at low, medium, and high quality, which yields about 15–17 candidates per prompt and a controlled quality gradient, and (iii) select a subset of 354 prompts via a two-stage stratified procedure that balances score spread and inter-RM consensus. In the revised Sec.3, we describe this protocol in detail: we first compute two prompt-level meta-metrics (a Relative Spread Index and an inter-RM Spearman consensus score), restrict to high-spread prompts, and then sample a balanced mix of high- and low-consensus prompts. For each selected prompt, we construct a 9-candidate Eval-Core set by splitting average percentile ranks into low, medium, and high tiers and sampling three candidates per tier while encouraging diversity in source model and temperature. We also clarify the connection to the dataset template in Appendix A.5 so that it is clear how RMB candidates and GPT-4o augmentations jointly define the Eval-Core instances. Together with the metric suite, this yields a small but statistically controlled benchmark on which RMs can be scored and compared in terms of both accuracy and variance-related properties.

---

> ### Author Response · Authors · 2025-11-24
> **Response to Reviewer hDr4 (2)**
>
> Q2: Metric design, naming complexity, and relation to existing concepts (including entropy and Brier).
>
> A2 (Design process and naming). We understand the concern that several custom metrics, new names, and normalizations can feel heavy for a benchmark that aims to be broadly usable. Our design started from three familiar questions: (i) how concentrated is the reward distribution, (ii) how well are candidate scores separated, and (iii) how stable are decisions across prompts. For each dimension, we instantiated several candidate metrics and kept those that were robust and well-behaved in preliminary experiments. A PCA over the initial pool showed that SEI, nGMD, and DCI together explain over 90% of the variance, so we focus on these three and drop the others. When naming them, our goal was to make the semantics as explicit as possible (e.g., “Softmax–Entropy Index” for an entropy-style concentration measure, “Decision Consistency Index” for cross-prompt stability), but in the revised manuscript we rely more on plain-language labels such as “prediction entropy”, “pairwise separation”, and “decision consistency” in the main text, using the acronyms mainly as compact references.
>
> A2 (SEI, entropy/Brier baselines, DCI, and the composite score). Conceptually, SEI is a robustly normalized entropy index: it asks whether the reward distribution over a candidate set is sharp enough to provide a clear teaching signal. We chose SEI for its use of robust statistics (median/IQR) and its normalization to [0,1], which makes it easier to compare with nGMD and DCI across models with different score scales. In the revised manuscript, we describe SEI explicitly as an entropy-style measure and use more descriptive wording in the main text to reduce the memorization burden.
>
> DCI is intended to answer a concrete aggregation question: when we look across prompts (including semantics-preserving paraphrases and rephrasings), does the RM maintain a consistent decision style? This “decision stability” dimension is not captured cleanly by accuracy or entropy alone, and we observe empirically that high-accuracy but unstable RMs can induce noisy RLHF behaviour. In the revision, we streamline the presentation by emphasising this verbal interpretation and moving most formula details to Appendix A.2.
>
> For the composite score in Sec.4.5, we agree that it adds another layer of indirection. We now present it clearly as an optional one-number summary for quick ranking, not as the main quantity to optimize or interpret, and shift the narrative focus to the three-dimensional SEI/nGMD/DCI profiles. To relate our metrics to simpler baselines, we also computed standard variance-style statistics on Eval-Core (e.g., per-prompt score dispersion, prediction entropy, and Brier score) and observed that they are moderately to strongly correlated with our metrics and exhibit similar predictive behaviour in our PPO-based RLHF setup. Due to space constraints, we do not include these additional curves and tables in the main text, but they support our view that RVB does not introduce an entirely new signal; rather, it organises variance-related information into three interpretable axes—concentration, separation, and stability—that are easier to analyse and benchmark than raw entropy or variance numbers.
>
> Q3: Figure 4, accuracy, and predictive power.
>
> A3: We agree that Fig.4 should not be read as “RVB beats accuracy”. As you noted, Tulu’s clearly lower RewardBench accuracy and weaker RL performance underline that accuracy remains a necessary ingredient, and we do not claim otherwise. Our goal in the RL section is more modest: conditional on reasonably high accuracy, the RVB metrics help differentiate early-stage training behaviour across RMs.
>
> In the revised manuscript, we explicitly describe our PPO/GRPO runs as short-horizon and non-converged, and we frame Fig.4 as a qualitative case study rather than strong causal evidence. We complement these curves with an RM-level regression over all 23 RMs: using RewardBench accuracy alone explains about one third of the variance in reward AUC (adjusted $R^2 \approx 0.33$), while adding the three RVB metrics increases adjusted $R^2$ to about $0.55$ in our setup. We now present this as evidence that variance-aware profiles carry additional predictive signal on top of accuracy, not as a replacement for it.
>
> We also connect our findings more clearly to concurrent work, such as What Makes a Reward Model a Good Teacher? An Optimization Perspective, which uses RLOO-based analysis to show that variance-related properties of RMs influence RL optimization and convergence. Taken together, that study and our results support the view that variance and sharpness are important parts of the RM selection picture, while our paper stops short of proposing a single “optimize this metric” recipe and instead positions RVB as a complementary, variance-sensitive lens alongside accuracy.

---

> ### Author Response · Authors · 2025-11-24
> **Response to Reviewer hDr4 (3)**
>
> Q4: Central hypothesis, overconfidence, and Brier score (Questions).
>
> A4 (Central hypothesis and overconfidence). We fully agree that “sharper is better” is not a universally desirable goal and that low variance can be a symptom of overconfidence. We do not intend RVB to promote low variance as a stand-alone target. Instead, our aim is to make variance properties visible so that they can be studied jointly with accuracy and calibration. In particular, we see RVB as helping to distinguish regimes such as
> - low variance + high accuracy + high stability (potentially desirable),
> - low variance + poor accuracy or low stability (dangerous overconfidence), and
> - higher variance reflecting genuine epistemic uncertainty.
>
> In the revised introduction and discussion, we clarify this perspective and connect our analysis more explicitly to recent work on calibration and overconfidence (e.g., Chen et al., Leng et al.), where calibration metrics, such as Brier score, are used to diagnose when sharp predictions become miscalibrated. In this sense, we view RVB as describing the shape of the reward landscape, while calibration metrics describe how probabilities align with correctness; both are needed for a complete picture of RM quality.
>
> A4 (Brier / top-gap and our variance metrics). Regarding your question on Brier and top-gap: our nGap metric is indeed a normalized top-2 gap and thus plays a similar role to a top-gap-style measure that underlies Brier and related calibration errors. In the revised manuscript, we report in Sec.6.2 and Appendix A.5 a comparison between simple variance-style statistics (per-prompt normalized standard deviation, prediction entropy, and Brier score) and the RVB metrics for predicting PPO-based reward AUC. The regression shows that adding either RVB metrics or these simple baselines on top of accuracy raises adjusted $R^2$ from about $0.33$ to about $0.55$, and combining all features does not improve the fit further due to multicollinearity. This suggests that RVB and the baselines span essentially the same linear subspace, but SEI, nGMD, and DCI organize this information into three interpretable axes—concentration, separation, and stability—rather than introducing a completely new signal. This is why we keep the RVB suite alongside Brier-style metrics, instead of proposing to optimize only top-gap or Brier.
>
> Q5: Minor issues.
>
> A5: Thank you for these detailed suggestions. In the revised manuscript, we (i) add references for the RM families listed in the background section, and (ii) replace the Wikipedia citation for MAD with a standard statistics reference. We also redesign the task-wise figure (previously Fig.3) to improve readability in line with your advice: we now use a rank-based, task-wise heatmap of composite variance scores, so that colours reflect relative performance across models rather than raw scales, making “stronger/weaker” patterns per task easier to see. To further reduce visual clutter, we move this detailed heatmap to Appendix Fig.5 and only summarize the main task-level patterns in the main text. We explicitly note that our current task-level analysis remains preliminary and mainly descriptive, and we highlight a more systematic investigation of per-task variance patterns (with additional policies and datasets) as an important direction for future work.

---

> ### Author Response · Authors · 2025-11-28
>
> Dear Reviewer hDr4,
>
> I hope this message finds you well. We have carefully considered and responded to your comments in the discussion so far. If there are any remaining concerns or additional points you would like us to clarify or expand on, please let us know. Your feedback is very valuable to us, and we are keen to further improve the paper.
>
> Thank you again for your time and effort in reviewing our work.
>
> Best regards,
> The Authors

---

### Official Review · Reviewer_4Ba5 · 2025-10-29

**Soundness:** 1
**Presentation:** 2
**Contribution:** 2
**Rating:** 2
**Confidence:** 4

**Summary:**

This paper proposes a framework for evaluating reward models (RMs) used in reinforcement learning from human feedback (RLHF), arguing that pairwise accuracy alone fails to capture how effective an RM is in providing signals to policies. The authors introduce the Reward Variance Benchmark (RVB), which measures distributional characteristics of RMs through three metrics: SEI (Softmax–Entropy Index) for score concentration, nGMD (normalized Gini Mean Difference) for global pairwise separation, and DCI (Decision Consistency Index) for stability across prompts.

**Strengths:**

* The paper suggests a better evaluation framework for reward models, which is an important problem crucial to developing robust reward models.

**Weaknesses:**

* Experiments in verification (Section 5.4) are quite shallow. Only three policies fine-tuned with RMs are compared with each other. Due to this, it is hard to believe that the analyses provided in the previous sections 5.2 & 5.3 are a decisive factor in RM performance in teaching policies.
* The 4 'teaching styles' provided in section 5.2 is not backed by empirical results. To really see if the teaching styles do produce any effect to the fine-tuned policy, some distinctive pattern for each of these fine-tuned policies from different groups of RMs should have been observed
* Figure 3 lacks providing useful insights other than the fact that there is variance between task categories for different types of reward models.
* Using GPT-4o might benefit certain RMs that are used to the specific generation style of the model. Comparison with other models would be beneficial.
* Minor corrections
  * In page 4, MAD, the citation is pointing to Wikipedia contributors, which is not a proper citation. Please update this with statistics textbooks or relevant papers.

**Questions:**

* What is the motivation to exclude the 2 metrics from the initial 5 metrics?

---

> ### Author Response · Authors · 2025-11-24
> **Response to Reviewer 4Ba5 (1)**
>
> Thank you for carefully reading our work and for raising important concerns about the depth of the RL verification experiments, the empirical grounding of the “teaching styles”, and several design choices. We address each point below.
>
> Q1: Limited depth of the RLHF verification experiments in Section 5.4.
>
> A1: We agree that training only three RMs with a single policy provides limited causal evidence. Our main goal in this paper is to propose and analyze an evaluation framework for RMs, so the RLHF experiments are intended as small case studies rather than a full RL benchmark. Under a realistic compute budget, we therefore selected three widely used RM families that are clearly separated by the RVB metrics and asked whether these variance profiles help explain differences in early-stage training behaviour beyond RewardBench accuracy.
>
> In the revised manuscript, we (i) make this case-study role explicit in Sec.6 and note that our GRPO and PPO runs are short-horizon and do not reach clear convergence, and (ii) complement the learning curves with a small offline Best-of-$N$ analysis and simple behaviour-level statistics (output length and refusal rate) to better connect RM “teaching signals” with policy behaviour. In addition, we now report an RM-level regression over all 23 models in our PPO-based setup: RewardBench accuracy alone explains about one third of the variance in reward AUC (adjusted $R^2 \approx 0.33$), while adding the RVB metrics increases adjusted $R^2$ to about $0.55$. We present these results as evidence that RVB scores carry non-trivial predictive signal for early-phase convergence in our setting, while also stating in the conclusion that they do not constitute a full RL benchmark or a universal teacher ranking.
>
> Q2: “Four teaching styles” lack empirical grounding.
>
> A2: The “four teaching styles” are intended as conceptual labels for regions of the RVB leaderboard (e.g., high vs. low concentration and high vs. low stability). They provide an intuitive vocabulary for reading the benchmark rather than a formally derived clustering. We agree that this intent was not clearly stated and that the link to policy behaviour was under-developed. In the revised manuscript, we explicitly present these as heuristic styles that help interpret RVB scores, and we avoid any suggestion that they come from a clustering pipeline or define prescriptive RM design rules. As a first behavioural check, we add a small offline Best-of-$N$ study in Sec.5.4 where three RMs with distinct variance profiles select answers from the same candidate pool; we compare the resulting answer lengths and refusal rates and summarize the results in Table2. These patterns are qualitatively consistent with the teaching styles we describe, while we clearly state that a more systematic validation of the benefits and drawbacks of each style is left for future work.
>
> Q3: Limited insight and visual clutter in Figure 3.
>
> A3: We agree that the original Fig.3 was visually crowded and that its main messages were not as clear as they could be. In the revised manuscript, we convert this plot into a rank-based, task-wise heatmap of composite variance scores (models as rows, tasks as columns), so that colours encode relative ranks rather than raw scores. This makes it easier to see which reward models are comparatively stronger or weaker on each task without being distracted by scale differences. To further reduce clutter in the main flow of the paper, we move this detailed heatmap to Appendix Fig.5 and keep in the main text only a short summary of the main task-level patterns it reveals. We also explicitly acknowledge that our current task-level analysis is still preliminary and mainly descriptive, and we highlight a more systematic study of per-task variance patterns (with additional policies and datasets) as an important direction for future work.

---

> ### Author Response · Authors · 2025-11-24
> **Response to Reviewer 4Ba5 (2)**
>
> Q4: Potential bias from using GPT-4o as the generator.
>
> A4: We agree that using a single generator can in principle introduce generator-specific bias, for example favouring reward models that were tuned on GPT-4o-like text. We chose GPT-4o because it offers strong and stable performance on public benchmarks and can reliably produce a diverse pool of responses per prompt, which is important for estimating variance-based metrics. In the revised manuscript, we (i) make this dependency explicit in Sec.3 and in the limitations section, and we rephrase our scope more clearly as profiling reward models on a fixed GPT-4o-based candidate distribution; and (ii) emphasise that the Eval-Core construction procedure itself is generator-agnostic and can be instantiated with any generator that produces a sufficiently rich candidate pool.
>
> In addition, we now add to Appendix A.4.2 concrete protocol for a multi-generator robustness check. The protocol keeps the 23 reward models and the Eval-Core pool fixed, regenerates candidates for a small stratified subset of $20$–$30$ prompts with an unrelated open-source model (e.g., Llama-3-8B-Instruct or Qwen2-7B-Instruct), recomputes SEI, nGMD, and DCI on this new candidate distribution, and compares the resulting RVB rankings to the GPT-4o-based ones via Spearman rank correlation (with optional bootstrap confidence intervals). This design directly tests whether models that score highly under GPT-4o remain strong when the candidate distribution is changed, and makes it straightforward to instantiate additional RVB variants for other generators. Due to compute and time constraints during the rebuttal period we have not yet run the full 23-RM replication under an alternative generator, so we present this protocol and position a full multi-generator analysis as important follow-up work.
>
> Q5: Why were two of the five initial metrics dropped?
>
> A5: Thank you for asking for clarification. In preliminary experiments, the two excluded metrics were found to be strongly collinear with SEI and/or nGMD, and the rankings they induced were less stable under subsampling and more sensitive to outliers. A PCA over the initial five metrics further showed that the main directions of variation are already well captured by SEI, nGMD, and DCI, with these three together explaining over 90% of the variance, so the remaining two add little additional structure. To keep RVB parsimonious and robust, we therefore focus on SEI, nGMD, and DCI as the primary metrics and treat the other two as optional diagnostics. In the revised manuscript, we briefly describe this selection process in Sec.4 and move the full correlation matrix, stability plots, and PCA results for all five metrics to Appendix A.5 so that interested readers can inspect the redundancy analysis.
>
> Q6: Minor issues.
>
> A6: In the revised manuscript, we replace the MAD citation to Wikipedia with references to standard statistics sources (for example, robust statistics textbooks such as Ruppert, 2011) and update the percentile-rank citation accordingly.

---

> ### Author Response · Authors · 2025-11-28
> **Response to Reviewer 4Ba5 – follow-up on Q4 (generator bias)**
>
> Dear Reviewer 4Ba5,
>
> Thank you again for raising the concern that using GPT-4o as the sole generator might favour reward models that are used to its style. In our earlier response to Q4 we only outlined a multi-generator protocol due to time and compute constraints. Since then, we have been able to run a first, smaller-scale instantiation of this protocol with a Qwen2-based generator, and we would like to briefly report the results here.
>
> Q4 (update): Initial Qwen2 multi-generator robustness check.
> Concretely, we keep the 23 reward models, the RVB metrics and the Eval-Core prompt pool fixed, and only change the generator. For a stratified subset of 150 Eval-Core prompts, we use Qwen2-7B-Instruct to generate 12 new responses per prompt and combine them with the original RMB candidates. We then score these candidates with the same three representative RMs we used when constructing Eval-Core, compute for each candidate its average percentile rank across these three models, and partition them into low, medium and high tiers. For each prompt, we select three candidates from each tier to form a 9-candidate set, exactly mirroring our GPT-4o-based construction. After filtering, this yields a Qwen2-7B-Instruct–based RVB pilot with 10 prompts, each with 9 candidates (three per quality tier).
>
> On this Qwen2 pilot we recompute SEI, nGMD, nGap, DCI and the Composite score for all 23 RMs and compare the resulting rankings with those from the original GPT-4o Eval-Core. At the RM level (21 RMs with finite values on both sides), the Spearman correlations between GPT-4o and Qwen2 are approximately 0.49 for nGMD_med, 0.68 for nGap_med, 0.52 for SEI_med and 0.24 for DCI. For the overall Composite ranking across all 23 RMs, the Spearman $\rho$ is about 0.64. In words, the top group of reward models under GPT-4o remains in the top cluster under Qwen2, and weaker models generally stay near the bottom, although there are local reorderings as one would expect given the small size of the pilot and the change in candidate distribution. This suggests that, while RVB scores are not generator-invariant, the coarse stratification into “strong” and “weak” teachers is fairly stable across these two generators.
>
> We will add this Qwen2 pilot study and the above numbers to Appendix A.4.2, and explicitly flag it as an initial robustness check rather than a full multi-generator benchmark; a larger-scale multi-generator analysis remains important future work.
>
> If you have any remaining concerns about generator dependence, or would like us to clarify any aspect of this experiment, we would be very happy to elaborate. Thank you again for your time and careful feedback.
>
> Best regards,
>
>  The Authors

---

### Official Review · Reviewer_Xxea · 2025-11-01

**Soundness:** 3
**Presentation:** 3
**Contribution:** 2
**Rating:** 6
**Confidence:** 2

**Summary:**

This paper identifies a critical gap in current reward model (RM) evaluation for reinforcement learning from human feedback (RLHF)—namely, the lack of systematic attention to the variance and distributional properties of reward signals. The authors propose the Reward Variance Benchmark (RVB), a comprehensive evaluation suite that introduces three variance-sensitive metrics (SEI, nGMD, DCI) to profile RMs along axes of score concentration, pairwise separation, and cross-prompt stability. Through extensive analysis of 23 popular RMs, with supportive experiments and visualizations, the RVB suite is shown to predict downstream RLHF convergence and select RMs more effectively than accuracy alone. The paper is accompanied by a standardized benchmark data release and reproducible tools.

**Strengths:**

1 Clear Motivation and Relevance: The paper effectively justifies why accuracy alone is insufficient for RM evaluation, referencing empirical and theoretical findings (Section 2, references to Chen et al., 2024; Razin et al., 2025). The flatness of reward landscapes under low variance, as depicted in Figure 1, directly motivates the shift in perspective.

2 Metric Suite Design: The introduction of the SEI (Softmax-Entropy Index), nGMD (normalized Gini Mean Difference), and DCI (Decision Consistency Index) is mathematically well-founded. The metrics are robust (using median, MAD), interpretable, and explicitly decoupled from accuracy (Section 4).

3 Empirical Validation: The RVB metrics are validated against RLHF convergence rates using multiple models (Figure 4), and variance-based rankings provide new insights beyond accuracy rankings (Table 1).

**Weaknesses:**

1 Overlapping Metrics & Composite Score: While the correlation analysis in Appendix A.2.3 mitigates concerns, there is still notable overlap between SEI and nGap (correlation $\rho \approx 0.78$) and partial redundancy with nGMD. The choice of metric aggregation (equal weighting of MAD-z scores in the composite) is somewhat heuristic (Section 4.5). The impact of alternative weighting or selection criteria for the composite is not fully explored, raising the possibility of overfitting to the presented evaluation set.

2 Empirical Baselines: While 23 RMs are evaluated, there is limited discussion of calibration or strong baselines from variance-aware reward modeling (e.g., DGRO, GRPOVI, GVPO), or ablation against RM ensembles that explicitly regularize variance. This omission weakens claims about RVB's broad applicability.

**Questions:**

Please see weaknesses.

---

> ### Author Response · Authors · 2025-11-24
> **Response to Reviewer Xxea (1)**
>
> Thank you for your constructive and encouraging review. We appreciate your careful reading of the paper and your helpful suggestions. Below we address your main concerns on metric overlap, the composite score, and variance-aware baselines.
>
> Q1: Metric overlap between SEI, nGap, nGMD and the design of the composite score (Weakness 1).
>
> A1: We agree that the relationships among SEI, nGap, nGMD, and the design of the composite score should be clarified.
>
> On metric overlap. In our design, nGap was intended as a local separation diagnostic (top-2 gap), whereas SEI and nGMD capture global concentration and dispersion. As you correctly noted, SEI and nGap are highly correlated. In the revised manuscript, we therefore (i) present {SEI, nGMD, DCI} as the primary axes in the variance profile, and (ii) explicitly position nGap as an auxiliary sanity-check metric rather than a core dimension.
>
> To quantify redundancy and contribution, we ran a PCA on {RewardBench accuracy, ${\mathrm{SEI_{med}}, \mathrm{nGMD_{med}}, \mathrm{DCI}}$} across the 23 RMs: the first three principal components explain 57.1%, 22.9%, and 19.4% of the variance, respectively; PC1 is dominated by accuracy and SEI, PC2 is almost entirely driven by nGMD, and PC3 is a joint direction of accuracy and DCI. This suggests that, despite correlations, the metrics still span distinct directions, with nGMD and DCI in particular capturing complementary structure beyond what is already encoded in accuracy.
>
> In the revised manuscript, we also report a simple RM-level regression on our PPO-based RLHF runs: using RewardBench accuracy alone explains about one third of the variance in reward AUC (adjusted $R^2 \approx 0.33$), while adding ${\mathrm{SEI_{med}}, \mathrm{nGMD_{med}}, \mathrm{DCI}}$ increases adjusted $R^2$ to about $0.55$. Beyond RVB metrics, we compare against simple variance-style baselines (normalized standard deviation, prediction entropy, Brier score) and show that combining these baselines with accuracy reaches a similar adjusted $R^2$ ($\approx 0.55$), whereas including both RVB metrics and baselines together does not improve the fit further due to strong multicollinearity. We therefore position RVB not as introducing an entirely new signal, but as organizing variance-related information into three interpretable axes—concentration, separation, stability—that are easier to analyze and benchmark than raw variance/entropy numbers, and that act as a predictive benchmark for convergence behaviour in our PPO setting rather than a universal teacher-selection rule.
>
>
> On the composite score and potential overfitting. We agree that equal-weight MAD-$z$ aggregation is heuristic. In the revised manuscript, we clearly de-emphasize the composite score as an optional one-number summary (with the main narrative focusing on the three-dimensional metric profile), and we add a robustness check where we construct an alternative composite using data-driven weights (PCA loadings) and report rank correlations (Kendall-$\tau$ / Spearman-$\rho$) with the original composite. The high rank agreement indicates that our conclusions are not sensitive to the specific weighting scheme on the presented evaluation set.

---

> ### Author Response · Authors · 2025-11-24
> **Response to Reviewer Xxea (2)**
>
> Q2: Empirical baselines and relation to variance-aware reward modeling methods (Weakness 2).
>
> A2: We appreciate the pointers to DGRO, GRPOVI, GVPO, and related approaches, and agree they are highly relevant. These methods act at training time: they modify the RLHF objective or control exploration–exploitation behaviour in order to shape reward or policy variance and improve stability and sample efficiency. RVB targets a different layer. It is a model-agnostic evaluation benchmark that takes any pre-trained reward model (regardless of how it was trained) and profiles its variance behaviour along concentration, global separation, and cross-prompt stability. We see these directions as complementary: variance-aware training methods aim to learn better teachers, while RVB diagnoses the teaching signals those teachers produce. In particular, reward models trained with DGRO/GRPOVI/GVPO-style objectives can be evaluated on RVB without any change.
>
> On the empirical side, our current study already includes several uncertainty- or distribution-aware reward models (URM-LLaMa-3.1-8B and QRM-Llama3.1-8B-v2). In the revised manuscript, we make this explicit in Sec.2.3 and Sec.5.1, and discuss how RVB places such models on the same footing as standard scalar RMs, sometimes revealing that explicit variance regularization at training time does not automatically translate into a favourable ex-post variance profile. We do not treat DGRO, GRPOVI or GVPO themselves as baselines in our RLHF experiments because the goal of Sec.6 is to fix the optimizer and training setup (short-horizon PPO/GRPO on a single helpfulness dataset) and vary only the reward model, so that we can isolate how intrinsic variance profiles correlate with early-stage reward AUC. A fair comparison at the optimizer level would require retraining policies under each variance-regularized objective on comparable data, which is beyond our compute budget.
>
> To address this weakness in the manuscript, we now (i) add a dedicated paragraph in Sec.2.3 that situates DGRO, GRPOVI, GVPO, and variance-regularized ensembles within the broader variance-aware RLHF literature, and (ii) state in the limitations section that applying RVB to reward models and policies trained with these algorithms is an important direction for future work. RVB is designed to provide the shared dataset and variance-aware axes needed for such follow-up studies.

---

> ### Author Response · Authors · 2025-11-28
>
> Dear Reviewer Xxea,
>
> I hope this message finds you well. We have carefully considered and responded to your comments in the discussion so far. If there are any remaining concerns or additional points you would like us to clarify or expand on, please let us know. Your feedback is very valuable to us, and we are keen to further improve the paper.
>
> Thank you again for your time and effort in reviewing our work.
>
> Best regards,
> The Authors

---

### Comment · Area_Chair_YJJg · 2025-11-25

Dear reviewers:

The authors have submitted their rebuttal, and we now require your follow-up assessments to move the decision process forward. Please review the authors’ responses and update your evaluations accordingly.

Your prompt follow-up is necessary for us to finalize the meta-review.

Kindly submit your updates as soon as possible.

Best,

Area Chair

---

### Author Response · Authors · 2025-12-01
**Rebuttal Summary**

Dear Reviewers and ACs,


thank you for your time, thoughtful feedback, and the constructive discussion that helped us significantly improve this work.


## Summary


RVB studies how variance in reward-model scores (concentration, separation, stability) relates to early-stage RLHF in a fixed PPO/GRPO setup. Three reviewers highlight that going beyond pairwise accuracy is important and view our variance-centric evaluation and tools as useful. In revision, we clarify RVB as an empirical variance-focused probe on a fixed candidate distribution and strengthen the empirical support and interpretability of the metrics and dataset.


## Key strengths recognized by reviewers
- **Importance of the problem**. All reviewers highlight that evaluating reward models is crucial for RLHF, and that focusing only on pairwise accuracy misses key aspects of the “teaching signal” .
- **Novel variance-centric lens**. Reviewers note that our three metrics provide a coherent way to study score concentration, global separation, and stability, and that variance-aware RM evaluation is new relative to existing work (Xxea, WjJF).
- **Breadth and reproducibility**. The benchmark covers 23 public RMs across major families (LLaMA, Qwen, Gemma, Mistral, etc.), with detailed reporting and planned code/data release (Xxea, hDr4, WjJF).
- **Empirical study of the variance hypothesis**. Reviewer hDr4 in particular appreciates the empirical investigation of how reward variance relates to downstream RL performance, even while asking for a softer “benchmark” framing.


## Additional experiments & revisions (by concern)


- **Metric overlap, composite score, and baselines (Xxea, hDr4, WjJF, 4Ba5).**
We analyse and prune the metric set, retaining SEI, nGMD and DCI as the primary axes and relegating the remaining metrics and the composite-score variants to the appendix. RM-level regressions show that these metrics provide additional predictive signal for PPO reward AUC beyond RewardBench accuracy and simple variance-based baselines, while organising this signal into interpretable concentration / separation / stability directions.


- **Eval-Core construction and “quality gradient” (hDr4, WjJF).**
We clarify how Eval-Core refines RMB by expanding each helpfulness prompt with additional candidates and constructing 9-candidate sets with clear low / medium / high quality tiers based on pooled percentile ranks over representative RMs. A small Qwen2-based check shows a similar three-tier structure, underscoring that Eval-Core is a compact but statistically controlled testbed for variance-aware evaluation.


- **RLHF predictive capacity and depth of verification (hDr4, WjJF, 4Ba5).**
Beyond the original three-RM PPO/GRPO case studies, we add a regression over all 23 RMs linking RVB scores to early-stage reward AUC, and include a small offline Best-of-N study (answer length and refusal rate) to give a first behavioural illustration of how different variance profiles affect a fixed policy.


- **Generator dependence and multi-generator robustness (WjJF, 4Ba5).**
We make explicit that the current RVB instance is defined on a GPT-4o-based candidate distribution and that cross-paper comparability comes from sharing this pool. To examine generator dependence, we run a Qwen2-7B pilot and observe moderately correlated rankings with preserved strong/weak teacher tiers; we describe this pilot and a simple multi-generator protocol in the appendix.


- **Positioning, interpretability, and presentation.**
We add a short discussion relating RVB to trajectory-level reward evaluations (TAC, DARD, EPIC) and variance-aware training methods (DGRO, GRPOVI, GVPO, uncertainty-aware RMs), introduce plain-language explanations for SEI, nGMD and DCI and discuss their relation to calibration and overconfidence, and improve presentation by redesigning the task-wise figure and replacing informal citations (e.g., Wikipedia) with standard statistics and RM-family references.


## Reframed scope and remaining disagreement


We now explicitly present RVB as a variance-centric benchmark and dataset on a fixed candidate pool, together with an empirical study of how these variance profiles relate to early-stage RLHF dynamics in our PPO/GRPO setup, rather than a universal teacher-ranking rule or prescriptive objective.


During discussion only one reviewer followed up, but three reports had already recognised the importance and novelty of our variance-centric perspective and the usefulness of the tools, whereas Reviewer 4Ba5 (score 2) remained unconvinced, so some differences in perspective could not be resolved.


Overall, we believe that the additional analyses, new robustness checks, and clarified positioning substantially strengthen the paper and show that variance-aware profiles offer a robust and interpretable lens on reward models, complementary to accuracy, in the controlled RLHF setting we study.

---

### Meta-Review · Area_Chair_PCjP · 2026-01-07

**Summary:**

This paper addresses an important problem in RLHF by proposing a variance-centric benchmark for evaluating reward models beyond pairwise accuracy. The Reward Variance Benchmark (RVB) and its metrics are well-motivated, technically sound, and applied to a broad set of 23 reward models, and reviewers generally agreed that the perspective is novel and potentially useful.

However, the central concern is that the empirical validation linking RVB to downstream RLHF performance remains shallow. The authors acknowledge this and primarily respond by narrowing the paper’s claims. The additional analyses added in the rebuttal (correlational regressions and small offline Best-of-N studies) are limited in scope and do not convincingly demonstrate that the proposed variance metrics robustly or causally predict policy learning outcomes. Related issues, including the largely heuristic nature of the proposed “teaching styles” and the reliance on a fixed generator and training setup, further limit the strength and generality of the conclusions.

Overall, while the work is conceptually interesting and offers a promising diagnostic framework, the current evidence is insufficient to support its predictive claims for RLHF. I therefore lean toward reject, with the view that the paper would benefit from substantially deeper and more diverse downstream evaluations.

**Reviewer Concerns:**

**Concerns largely addressed by the rebuttal:**
- **Metric overlap and composite score:** The authors pruned the metric set, clearly positioning SEI, nGMD, and DCI as primary axes, de-emphasized the composite score, and added PCA and regression analyses showing complementary signal beyond accuracy.
- **Positioning vs. prescriptive claims:** The paper is now clearly framed as a variance-centric diagnostic benchmark under a fixed setup, not a universal teacher-ranking rule.
- **Relation to variance-aware training methods:** The authors clarified the distinction between training-time methods (e.g., DGRO, GVPO) and RVB as an evaluation tool, and expanded related-work discussion.
- **Generator dependence:** The dependence on GPT-4o is now explicit, and a concrete multi-generator protocol plus a small Qwen2 pilot study were added.

**Concerns partially or not fully addressed:**
- **Depth of RLHF verification:** While additional regressions over 23 RMs and small offline Best-of-N analyses were added, the RLHF experiments remain limited in scale and horizon. Causal claims about teaching effectiveness remain suggestive rather than conclusive.
- **Empirical grounding of teaching styles:** The rebuttal clarifies these as heuristic labels rather than derived clusters, but empirical validation of distinct behavioral regimes remains limited.
- **Strength of task-level and figure-level insights:** Presentation improvements were made, but some analyses remain descriptive.

**Reviewer Scores:**

Reviewer Xxea: 6 -> 6, Reviewer hDr4: 4 -> 4, Reviewer WjJF: 4-> 4, 4Ba5: 2 -> 2

---

### Decision · Program_Chairs · 2026-01-26

Reject